# Constraining the equation of state in neutron-star cores via the long-ringdown signal

Christian Ecker [1] ✉, Tyler Gorda [1,2,3] ✉, Aleksi Kurkela [4] ✉ & Luciano Rezzolla [1,5,6] ✉

Multimessenger signals from binary neutron star (BNS) mergers are promising tools to infer the properties of nuclear matter at densities inaccessible to laboratory experiments. Gravitational waves (GWs) from BNS merger remnants can constrain the neutron-star equation of state (EOS) complementing constraints from late inspiral, direct mass-radius measurements, and ab-initio calculations. We perform a series of general-relativistic simulations of BNS systems with EOSs constructed to comprehensively cover the high-density regime. We identify a tight correlation between the ratio of the energy and angular-momentum losses in the late-time portion of the post-merger signal, called the long ringdown, and the EOS at the highest pressures and densities in neutron-star cores. Applying this correlation to post-merger GW signals significantly reduces EOS uncertainty at densities several times the nuclear saturation density, where no direct constraints are currently available. Hence, the long ringdown can provide stringent constraints on material properties of neutron stars cores.

The densest matter in the observable universe is found in the cores of neutron stars (NSs), where gravity compresses it to supernuclear densities, exceeding manyfold the nuclear density of $n_{sat} = 0.16$ baryons/fm$^3$. While the behaviour of pressure and density in such cores, that is, the equation of state (EOS) of strongly interacting matter, remains an open question, a precise determination of the EOS of NSs would provide precious insights on the phase diagram of Quantum Chromodynamics (QCD).

There have been remarkable advances in the inference of the EOS thanks to rapidly advancing observations of NSs and theoretical ab-initio calculations (see, e.g., ref. [1]). In addition, the observation of the GW signal from the late inspiral of the binary neutron star (BNS) merger event GW170817 demonstrated the potential of GW measurements to constrain the EOS by setting limits on the tidal deformabilities of the inspiralling NSs, which are tightly correlated with the EOS at densities around 3 $n_{sat}$ and reached by NSs prior to merger (see, e.g., refs. [2–4] for reviews). Finally, the analysis of the electromagnetic counterpart associated with GW170817 provided convincing evidence for the formation of a hypermassive neutron star (HMNS) that collapsed into a black hole over a timescale that, under a number of assumptions, has been estimated to be of approximately one second after the merger[5,6]. Because HMNSs are expected to reach densities that are significantly higher than those in the stars before the merger, the study of their properties opens up the opportunity to directly determine the EOS up to the highest densities in the observable Universe.

This potential will be fully realized with the upcoming third-generation GW observatories[7,8], whose high sensitivity at frequencies larger than 1 kHz allows them to detect with high signal-to-noise ratios (SNR) also the post-merger signal from the HMNS[2–4]. This has

[1]Institut für Theoretische Physik, Goethe Universität, Frankfurt am Main, Germany. [2]ExtreMe Matter Institute EMMI, GSI Helmholtzzentrum für Schwerionenforschung GmbH, Darmstadt, Germany. [3]Department of Physics, Technische Universität Darmstadt, Darmstadt, Germany. [4]Faculty of Science and Technology, University of Stavanger, Stavanger, Norway. [5]Frankfurt Institute for Advanced Studies, Frankfurt, Germany. [6]School of Mathematics Trinity College, Dublin, Ireland. ✉e-mail: ecker@itp.uni-frankfurt.de; gorda@itp.uni-frankfurt.de; aleksi.kurkela@uis.no; rezzolla@itp.uni-frankfurt.de

important consequences, as a number of studies have shown that the most prominent features of the power spectral density (PSD) of the post-merger signal, correlate with the underlying EOS models (see, e.g., refs. 9–14).

We here propose an improvement on this approach that concentrates on a specific and late part of the post-merger signal and that allows for a direct determination of the EOS at the highest densities. More specifically, just like the ringdown of a perturbed black hole contains precise information on the black hole properties, we show that the late-time, attenuated GW signal produced by the HMNSs between approximately 1 and 15 ms holds a similar potential in showing a clear correlation with the maximum densities and pressures of the EOS. We refer to this late-time signal as to the long ringdown since its characteristic damping time is much longer than the typical damping time of a black hole with similar mass. The origin of this long ringdown is to be found in the fact that approximately 10 ms after the merger—when the GW amplitude is still comparatively large—the HMNS exhibits a quasi-stationary dynamics, with a mostly axisymmetric equilibrium and small deformations with spherical harmonic degree $\ell = 2$ and azimuthal order $m = 2$ that are responsible for an almost constant-frequency, constant-amplitude GW emission. Under these conditions, a simple toy model, as that introduced in ref. 10, is sufficient to show that thanks to the equilibrium achieved during this stage, the radiated GW energy and angular momentum follow a linear relation and correlate strongly with the EOS at high densities. Hence, the observation of the long ringdown at a large SNR represents a faithful probe of the largest densities and pressures of the remnant's EOS.

## Results

### EOS properties

In order to quantify this correlation, we perform a suite of general-relativistic simulations of BNS mergers with EOSs carefully constructed so as to comprehensively cover the currently allowed space of parameters. More precisely, we employ a large posterior sample of model-agnostic, Gaussian-process (GP) based, zero-temperature EOSs of NS matter from[15] that is conditioned with constraints from the tidal-deformability measurement of GW170817, radio measurements of high-mass pulsars, combined mass-radius measurements from X-ray pulse-profile modeling of isolated NSs, as well as with low-energy nuclear-theory constraints from chiral effective field theory (CEFT) and high-energy particle-theory bounds from perturbative QCD. The significant breadth of EOSs in the ensemble reflects the current level of

uncertainty in the determination of the EOS. Because it is computationally prohibitively expensive to scan a large number of EOSs, we reduce the full ensemble to a smaller sample of six golden EOSs that maximizes the variation in the following four NS parameters: the maximum mass—also known as the Tolman-Oppenheimer-Volkoff (TOV) mass—of an isolated, nonrotating NS $M_{\rm TOV}$, its dimensionless compactness $C_{\rm TOV} := M_{\rm TOV}/R_{\rm TOV}$, where $R_{\rm TOV}$ is the corresponding radius, the central pressure $p_{\rm TOV}$, and the radius of a typical $1.4\,M_\odot$ NS $R_{1.4}$. By performing a principal-component analysis (see Methods section for details), we select six EOSs that are located in the center (EOS labeled F) and distributed on the boundary (EOSs labelled A–E) of the 68%-credible region in the four-dimensional space spanned by the NS parameters. We have chosen this region so that our sample characterises the distribution where most of the posterior weight is; a different choice of, e.g., 95% would consist of EOSs that are already in tension with observations and would not necessarily characterise the distribution as faithfully as the sample would be sensitive to the tails of the distribution.

These six EOSs are shown in Fig. 1 in the pressure–number density $(p, n)$ plane (left panel), along with the corresponding mass-radius relationships for nonrotating stars (right panel). We note that, by construction, our global sample of EOSs, and hence also the golden EOSs, do not contain strong phase transitions, which could lead to a larger EOS space consistent with astrophysical bounds[16]. This choice allows us to focus our attention on smooth EOSs and to build an understanding of their phenomenology, leaving the exploration of EOSs with phase transitions to a subsequent work where we will employ the approach in refs. 17,18.

While NSs during the inspiral stage can be described also when neglecting the temperature dependence of the EOS, during and after the merger shock-heating effects lead to non-negligible temperatures inside the merger remnant. To model and approximate these heating effects, the zero-temperature EOSs selected in our sample are modified in what is normally referred to as a hybrid EOS, where the cold part of the EOS is combined with an ideal-gas EOS, thus providing an effective-temperature contribution. While this is an approximation, it does not affect the properties of the correlation and we have adopted an adiabatic index value of $\Gamma_{\rm th} = 1.75$, which is close to the optimal value $(\Gamma_{\rm th} = 1.7)$ suggested in ref. 19 and on average mimics well the temperature dependence of microscopic constructions[20] [see the Supplementary Information (SI) for details]. We have verified that all of the qualitative properties of the GW emission from the HMNS presented here are preserved when considering also self-consistent temperature-

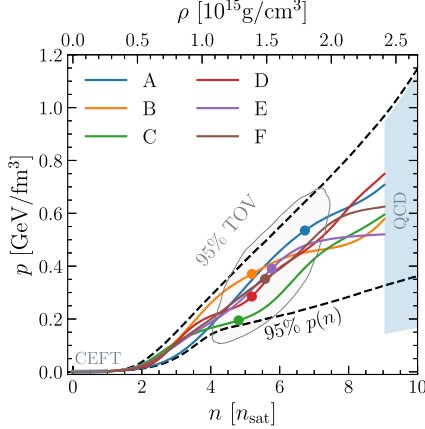
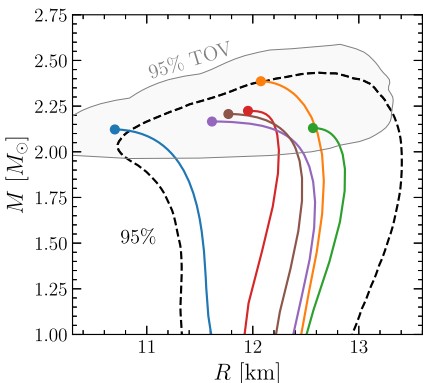

**Fig. 1 | The golden EOSs.** *Left panel:* Solid lines of different colors show the six golden EOSs (A–F) in the $(p, n)$ plane. The dashed black lines show the 95% credible intervals of all possible EOSs, while the CEFT and QCD bounds are shown with shaded areas (gray and light blue, respectively). Colored filled circles show the TOV points of the golden EOSs, while the solid light gray line is the 95% credible interval for all TOVs. *Right panel:* The same as in the left but shown in the $(M, R)$ plane. Source data for this figure are provided as a Source Data file.

dependent EOSs, such as the holographic Veneziano-QCD model (V-QCD)[20] or the Hempel-Schaffner-Bielich modified density-dependent model (HS-DD2) EOS[21] (see SI).

## Merger simulations

Using these six golden EOSs, we performed a series of general-relativistic BNS merger simulations and extracted the emitted GW signal starting from about 15 ms before merger until 30 ms after. From these simulations, we compute the instantaneous GW frequency $f_{GW}$, the radiated energy $E_{GW}$, and angular momentum $J_{GW}$. The binaries have been constructed with parameters that are consistent with those measured for GW170817, i.e., with fixed chirp mass $\mathcal{M}_{chirp} = 1.18\,M_\odot$ and three different ratios $q := M_2/M_1 = 0.7, 0.85, 1$ of the binary constituent masses $M_1$ and $M_2$. From a qualitative point of view, the dynamics of the six binaries reflects what has been found by a large number of works (e.g., refs. 10,12,22–27), with an HMNS attaining a metastable equilibrium a few milliseconds after the merger and then emitting GW radiation at frequency that is almost constant in time and around the characteristic $f_2$ frequency of the post-merger PSD[11]. Here, we instead focus on the rates at which energy and angular momentum are radiated by the HMNS when it has reached a quasi-stationary equilibrium at about 10 ms after the merger (see also refs. 28,29).

Figure 2 displays the most salient results of the six equal-mass binaries by showing in the top part the evolution of the radiated GW energy $E_{GW}$ and angular momentum $J_{GW}$ normalized by their values from the start of the simulations till the merger. (Note that the specific starting time of the simulation, or the time when the signal enters the detector's sensitivity curve, are not relevant here, since the energy and angular momentum radiated in the entire inspiral phase are negligibly small compared to the losses during and after the merger[30,31].) $E_{GW}^{mer}$ and $J_{GW}^{mer}$, where, as customary, $t_{mer}$ is defined as the moment of maximum-amplitude strain. Note the significant change in the evolution between the inspiral phase and the post-merger. In the former, all binaries—indicated with the same color convention as in Fig. 1—follow essentially the same trajectory in the $(E_{GW}, J_{GW})$ plane, which is well captured by the post-Newtonian (PN) approximation shown as a black dashed line (we use a reference tidal deformability $\tilde{\Lambda} = 580$ in the 5PN-order Taylor-T2 model of the PyCBC library[32]). Obviously, there are differences in evolution of the six binaries that are generated by the different tidal deformabilities, but these differences are minute when compared with those that emerge after the merger, when the different evolutions become visibly distinct. More importantly, it is remarkable that in the latter part of the signal, i.e., in the long ringdown, the normalized radiative losses in $E_{GW}$ and $J_{GW}$ are linearly related, as noted in refs. 28,29 and clearly shown in the inset reporting a magnification of the long ringdown. This apparently striking behaviour has a rather simple explanation in terms of the Newtonian quadrupole formula applied to a rotating system with an $\ell = 2 = m$ deformation, as in the toy model of Ref. 10. In this case, one can show the identity $\dot{E}_{GW}/\dot{J}_{GW} = dE_{GW}/dJ_{GW} = f_{GW}/(2\pi)$, where we use a dot to indicate a time derivative and $f_{GW}$ is the instantaneous GW frequency (for details see the Methods). Stated differently, during the long ringdown, the HMNS behaves essentially as a rotating $m = 2$ deformed quadrupole and radiates GW energy and angular momentum that are linearly related. Indeed, we have verified that more than 97% of the gravitational wave amplitude arises from this dominant mode. We note that while Fig. 2 refers to equal-mass binaries, a perfectly analogous behaviour is also realized by unequal-mass binaries (see Supplementary Fig. 5).

The importance of the results summarised in Fig. 2 is that binaries whose HMNS signal can be measured with high SNR, and hence for which the radiated GW energy and angular momentum can be measured more accurately, offer a key to access the properties of the EOS at the highest densities and pressures, i.e., $n_{TOV}$ and $p_{TOV}$. Before discussing how this can be done and to what precision, we need to make a few important remarks. First, the physical picture presented in Fig. 2 in terms of the slope between $E_{GW}$ and $J_{GW}$ can also be drawn in terms of the instantaneous GW frequency $f_{GW}$, which is instead presented in the bottom part of Fig. 2. This panel shows in fact that during the long ringdown $f_{GW}$ also asymptotes to an essentially constant value, $f_{GW} = f_{rd} \simeq$ const. (see also ref. 28 where this behaviour was also mentioned), and—assuming that the signal is dominated by the $\ell = 2 = m$ mode—this is the same value that can be deduced from the slope between $E_{GW}$ and $J_{GW}$. Hence, while measuring $f_{rd}$ in the long ringdown is conceptually analogous to measuring the slope, we have found that the latter is more robust as it is easier to fit an approximately linear function, i.e., $E_{GW}$ vs $J_{GW}$, than the average of an oscillating and potentially noisy function, i.e., $f_{GW}$. (This is because the

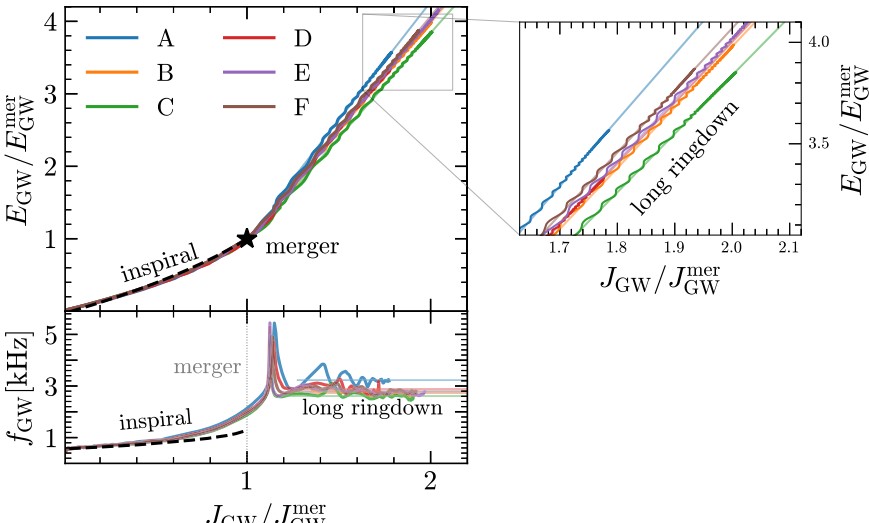

**Fig. 2 | Radiated GW energy and angular momentum.** *Top panel:* Using the same color convention as in Fig. 1, we report the energy $E_{GW}$ and angular momentum $J_{GW}$ emitted in GWs for the golden EOSs when normalized to the values at merger, indicated by a star. The evolution expected from the post-Newtonian inspiral is shown with a dashed black line. The inset offers a magnification of the long ringdown with the corresponding linear slopes indicated by thin lines of the same color. *Lower panel:* As in the top panel but in terms of the instantaneous GW frequency $f_{GW}$. The data refers to equal-mass binaries but very similar behaviour is found also for unequal-mass binaries (see Supplementary Fig. 5 for details). Source data for this figure are provided as a Source Data file.

variance of the slope in a linear fit of data $\{(x_i, y_i)\}$ is suppressed by an additional factor $\sum_i (x_i - \langle x \rangle)^2$ when compared to the residual variance of the fit.) The extrapolated slopes are shown with thin lines of the corresponding color in Fig. 2. Second, the long-ringdown frequency $f_{\rm rd}$ is close to but different from the main-peak frequency in the post-merger PSD, i.e., $f_2$, which is traditionally advocated as a good proxy for the EOS[9–13,33]. This is because $f_{\rm GW}$ oscillates wildly right after the merger and hence $f_2$ collects power from frequencies that are both larger and smaller than $f_{\rm rd}$ (see lower part of Fig. 2), thus increasing the uncertainty in its measurement. Stated differently, the difference between $f_2$ and $f_{\rm rd}$ is that the former collects power over a very broad window in time starting from the merger, while the latter contains information during a very narrow window around the long ringdown. Finally, our analysis reveals that the correlations between the GW signatures (i.e., $f_2$ and $f_{\rm rd}$) and the properties of the EOS (i.e., $n_{\rm TOV}$ and $p_{\rm TOV}$) are statistically different. In particular, we have measured the Pearson-correlation coefficients $r(X, Y) := \mathrm{cov}(X, Y)/(\sigma_X \sigma_Y)$ between the data from our golden EOSs to be $r(dE_{\rm GW}/dJ_{\rm GW}, p_{\rm TOV}) = 0.877$ and $r(dE_{\rm GW}/dJ_{\rm GW}, n_{\rm TOV}) = 0.917$ in the case of the slope, and $r(f_2, p_{\rm TOV}) = 0.792$ and $r(f_2, n_{\rm TOV}) = 0.865$ in the case of $f_2$, thus indicating that there is a strong correlation in both cases, but also that this is stronger for the long-ringdown frequency.

## Long ringdown in EOS inference

To illustrate how to make use of the long ringdown to set constraints on the EOS at the highest densities and pressures, Fig. 3 shows the tight correlation between the normalized slope during the long ringdown $d\hat{E}_{\rm GW}/d\hat{J}_{\rm GW}$, where $\hat{E}_{\rm GW} := E_{\rm GW}/E_{\rm GW}^{\rm mer}$ and $\hat{J}_{\rm GW} := J_{\rm GW}/J_{\rm GW}^{\rm mer}$, and the highest pressure $p_{\rm TOV}$ and density $n_{\rm TOV}$ reached in nonrotating NSs. The six different panels of Fig. 3 are organized so as to show in the three columns the correlations for the three different mass ratios considered ($q = 1$, 0.85, and 0.70 from left to right) and in the two rows the variation with the maximum pressure (top row) and the maximum density (bottom row). It is straightforward to appreciate from the six panels that the correlation is strong and we find it quite striking that

measuring the long-ringdown slope of a low-mass NS can provide precise information on the properties of matter at the highest densities and pressures realized in nature and which are well above those probed in the merger remnant.

To quantify the strength of the correlation, we consider a bilinear model fit to the data from our simulations

$$\frac{d\hat{E}_{\rm GW}}{d\hat{J}_{\rm GW}} = \beta_0 + \beta_1\, p_{\rm TOV} + \beta_2\, n_{\rm TOV} + \beta_3\, q + \beta_4\, q\, p_{\rm TOV} \\ + \beta_5\, q\, n_{\rm TOV} + \beta_6\, p_{\rm TOV}\, n_{\rm TOV}\,, \tag{1}$$

with model parameters $\beta_i$. Note that the ansatz (1) ignores quadratic terms in $q$, $p_{\rm TOV}$, and $n_{\rm TOV}$ as these provide only marginal improvements to the fit (for $q$) or break it (for $p_{\rm TOV}$, $n_{\rm TOV}$). After fitting this model over the time window $t - t_{\rm mer} \in [1, 30]$ ms, we obtain a probability distribution for the model parameters $\beta_i$ and, in turn, for the long-ringdown slope given the parameters of the EOS $P(d\hat{E}_{\rm GW}/d\hat{J}_{\rm GW}|p_{\rm TOV}, n_{\rm TOV}, q)$, which we use to produce the 68% (95%) credible intervals for $d\hat{E}_{\rm GW}/d\hat{J}_{\rm GW}$ shown in dark (light) shading in each of the panels in Fig. 3. These intervals are produced by marginalizing over the other EOS variables using the underlying probability distribution $P(p_{\rm TOV}, n_{\rm TOV})$ from the EOS ensemble.

Clearly, the bilinear model (1) reproduces well long-ringdown slopes, where the distribution of model parameters $\boldsymbol{\beta} := (\beta_0, ..., \beta_6)$ are given by a multivariate Gaussian distribution with a mean $\bar{\boldsymbol{\beta}} = (1.78, 0.72, -1.44, 1.90, -1.74, -1.14, 3.61)$ and a covariance matrix $\mathrm{cov}(\boldsymbol{\beta})$ reported in Supplementary Table 2, with $p_{\rm TOV}$ and $n_{\rm TOV}$ expressed in units of GeV/fm$^3$ and fm$^{-3}$, respectively.

Having pointed out the correlation between the properties of the long-term GW signal and the properties of the EOS at the *highest density*, we now take our analysis a step further and show how our results, in conjunction with a future post-merger GW detection, can be used to constrain the EOS at *all densities*. More specifically, to demonstrate the effect of a future measurement of $d\hat{E}_{\rm GW}/d\hat{J}_{\rm GW}$, we use

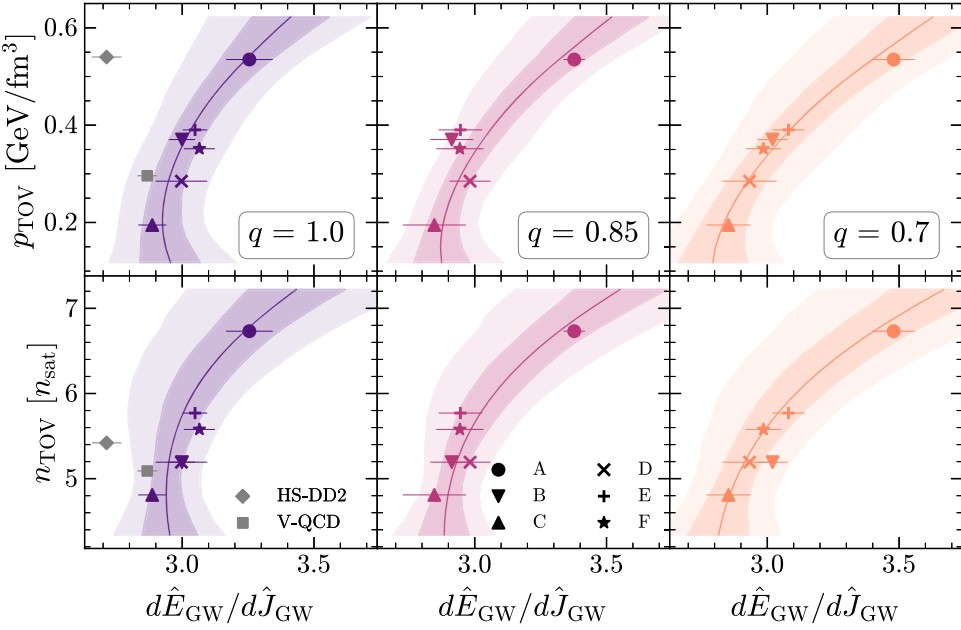

**Fig. 3 | Correlation between the long-ringdown slope and the TOV properties.** Data and fits illustrating the correlations between the slope of the radiated quantities $\hat{E}_{\rm GW}$ and $\hat{J}_{\rm GW}$ normalized to the merger values, and $p_{\rm TOV}$ (top row) or $n_{\rm TOV}$ (bottom row), and for different mass ratios (different columns). The dark (light) shaded regions denote 68%(95%) credible intervals for the bilinear model (1), while the solid lines denote the mean value. A marginalization over the remaining EOS quantity in the bilinear model was performed. Also shown are data points for two microscopic EOS models, one of which (HS-DD2) is disfavored by astrophysical data. Source data for this figure are provided as a Source Data file.

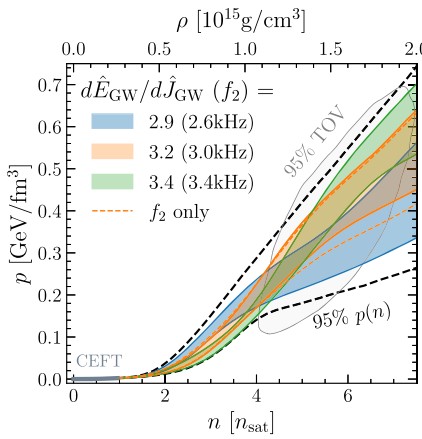
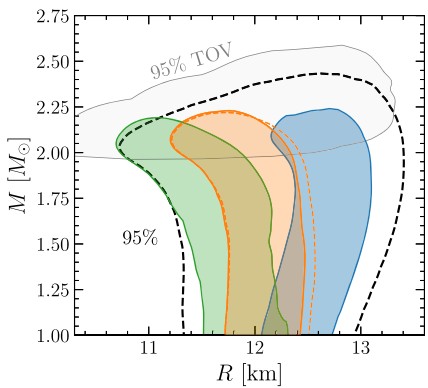

**Fig. 4 | Impact of slope measurements on the EOS and NS properties.** *Left panel* Using the same convention as in Fig. 1, we show the 68% credibility values on the $(p, n)$ plane from potential *joint* measurements of the long-ringdown slope and $f_2$ assuming a flat distribution for $q \in [0.7, 1]$ and an uncertainty of $\pm 4\%$ on the slope

and $\pm 4\%$ on $f_2$. Shown with a dashed line is the result for using only the $f_2$ measurement. *Right panel* the same but in the $(M, R)$ plane. Source data for this figure are provided as a Source Data file.

Bayes's theorem to infer the combined EOS and NS properties

$$P(\text{EOS}, \text{NSs}|\text{data}) = \frac{P(\text{data}|\text{EOS}, \text{NSs})P(\text{EOS}, \text{NSs})}{P(\text{data})} . \qquad (2)$$

In particular, we can use as an additional piece of the likelihood function $P(\text{data}|\text{EOS}, \text{NSs})$ the integral of our bilinear model over the likelihood of the measurement (see SI note 4 for the details). Assuming a Gaussian measurement of the long-ringdown slope, as well as a uniform distribution on the mass ratio $q \in [0.7, 1.0]$ from the measurement, we display in Fig. 4 the resulting constraints at 68% credibility.

The left panel of Fig. 4 considers three values of the measured long-ringdown slope, i.e., $d\hat{E}_{\text{GW}}/d\hat{J}_{\text{GW}} = 2.9, 3.2$ and $3.4$ with an error estimate of 8%, *joint* with the measurements of the frequency $f_2 = 2.6, 3.0, 3.4$ kHz with an error of 8%[33,34]. Considering that the standard deviation of the measurement of $d\hat{E}_{\text{GW}}/d\hat{J}_{\text{GW}}$ has been found to be of about 3%, our error estimates in Fig. 4 have been rather conservative. Only when using a distinct and more extensive analysis taking into account realistic GW-signal-processing pipelines will it be possible to set less conservative error estimates. For this analysis, we use a second bilinear model to fit the $f_2$ data, whose quality is found to be as good as the above fit to $d\hat{E}_{\text{GW}}/d\hat{J}_{\text{GW}}$. Using different colors for the different measurements, it is then apparent that smaller (larger) values of the slope would constrain the EOS to have higher (lower) pressures at 2–4 times nuclear saturation density while also having a smaller (larger) pressure and density for the maximally massive NSs. In turn, this leads to larger (smaller) radii for the most massive stars, as shown in the right panel of Fig. 4. Note also how the measurement of the long ringdown provides an improvement of the corresponding posteriors obtained when measuring the $f_2$ frequency only (see dashed orange lines for the representative case of $d\hat{E}_{\text{GW}}/d\hat{J}_{\text{GW}} = 3.2, f_2 = 3.2$ *kHz*). We should note that similarly large confidence intervals appear if we were to consider information on the slope only. Hence, Fig. 4 clearly highlights how the combination of information on the slope and on the $f_2$ frequency yields an increased accuracy in the properties of the EOS.

## Discussion

Our concluding remarks are about the robustness of the correlation. We have already commented that the results apply qualitatively unchanged when considering unequal-mass binaries or EOSs with a consistent temperature dependence (see SI note 5 for details). In addition, we have also verified that the same is true when considering different values for the chirp mass. More specifically, taking $\mathcal{M}_{\text{chirp}} = 1.13, 1.22\, M_\odot$ instead of our fiducial value of 1.18 leads to differences in the slope that are

significantly smaller than those introduced by considering different EOSs (see SI note 5 for details). Finally, and importantly, the long-ringdown slope is essentially insensitive to different choices in the adiabatic index $\Gamma_{\text{th}}$, as we have verified by replacing our fiducial value of 1.75 with $\Gamma_{\text{th}} = 1.5$ or 2 (see SI note 5 for details).

The preliminary study carried out here can be improved in a number of ways, e.g., by estimating the impact that large spins, strong magnetic fields, neutrino emission, strong first-order phase transitions, and temperature-dependent EOSs and more generic treatments of the crust and sub-saturation density matter have on the long-ringdown slope. Additionally, the set of neutron-star parameters used in the principal-component analysis could be extended and optimized. However, already now our correlation between the radiated energy and angular momentum during the long ringdown has the realistic potential of significantly reducing the EOS uncertainty at the highest densities realized in NSs for which no alternative observational constraints are available to date. This potential may already be exploited by the ongoing and near-future observations by the LIGO-Virgo-Kagra collaboration, but it will surely play a fundamental role in third-generation GW detectors, where the combined network of Cosmic Explorer and Einstein Telescope are expected to detect 180 BNS signals per year with post-merger SNR > 8[8].

## Methods

In what follows we provide additional details on a number of aspects of our analysis that we have omitted in the main text for compactness. These refer to the approach followed for the selection of the golden EOSs, the numerical techniques employed to simulate the binaries and extract the GW signal, and a number of validations highlighting the robustness of the correlation found between the EOS and the long-ringdown slope.

### Selection of the golden EOSs

For the agnostic construction of cold EOSs, we begin from the GP setup presented in[15], which we briefly review here. Below densities of $n = 0.57\, n_{\text{sat}}$, we use the crust model by Baym, Pethick, and Sutherland[35]; note that the uncertainty associated with the crustal EOS has been recently discussed in[36]. Above this density, in the interval $n = [0.57, 10]\, n_{\text{sat}}$, a GP regression is performed in an auxiliary variable $\phi(n) := -\ln(1/c_s^2(n) - 1)$, where $c_s$ is the sound speed, and where the prior for $\phi(n)$ is drawn from a multivariate Gaussian distribution

$$\phi(n) \sim \mathcal{N}\left(-\ln(1/\bar{c}_s^2 - 1), K(n, n')\right), \qquad (3)$$

with a Gaussian kernel $K(n, n') = \eta \exp(-(n - n')^2/2\ell^2)$[15]. The hyper-parameters $\eta$, $\ell$, and $\bar{c}_s^2$ within these definitions are themselves drawn from probability distributions

$$\eta \sim \mathcal{N}\left(1.25, 0.25^2\right), \quad \ell \sim \mathcal{N}\left(1.0\, n_{sat}, (0.2 n_{sat})^2\right),$$
$$\bar{c}_s^2 \sim \mathcal{N}\left(0.5, 0.25^2\right). \tag{4}$$

Below a density of $1.1\, n_{sat}$, the GP is conditioned with the CEFT results from[37]. In particular, the average between the soft and stiff results from that work are taken as the mean, while the difference between them is taken as the 90% credible interval for the conditioning[15]. From this GP, we draw sample of 120,000 EOSs.

We impose the astrophysical observations referred to as Pulsars + $\tilde{\Lambda}$ in[15]. Explicitly, we use the following three sets of observations:

1. heavy-pulsar mass constraints from radio astronomy. In particular, we use the constraints from PSR J0348 + 0432 with $M = 2.01 \pm 0.04\, M_\odot$[38] and PSR J1624 − 2230 with $M = 1.928 \pm 0.017\, M_\odot$[39]. We approximate the uncertainties from these measurements as normal distributions, which holds to good accuracy.
2. joint tidal-deformability and mass-ratio constraints from GW observations. We use the two-dimensional (2D) joint probability distribution for the tidal deformability and mass ratio using the `PhenomPNRT` waveform model with low-spin priors from Fig. 12 of ref. [40].
3. joint mass-radius measurements from X-ray pulse profile modeling. We use the 2D joint probability distribution for the mass and radius for PSR J0740 + 6620 using the NICER + XMM-Newton data from the right panel of Fig. 1 of ref. [41].

Within each of these measurements, we assume as our mass prior $P_0(M|\text{EOS})$ a uniform distribution between $0.5\, M_\odot$ and $M_{TOV}$.

Lastly, the GP is conditioned using information from high-density perturbative QCD calculations, which are under theoretical control at densities with baryon number chemical potential $\mu = 2.6$ GeV, corresponding to $n$ approximately $40 n_{sat}$. This information is included in a conservative way, excluding those EOSs which cannot be connected to the perturbative densities using any causal, mechanically stable, and thermodynamically consistent interpolation in the density interval $[10, 40]\, n_{sat}$[42]. This is done by conditioning the GP with the QCD likelihood function of ref. [15], where the uncertainty in the pQCD calculation at $\mu = 2.6$ GeV is taken into account by marginalizing over the unphysical renormalization scale $X := 2\Sigma/(3\mu)$ in the range $[0.5, 2]$, with $\Sigma$ the renormalization scale in the modified minimal subtraction scheme.

We consider the posterior in the 4D space of $(M_{TOV}, C_{TOV}, \ln p_{TOV}, R_{1.4})$, within which we perform a modified principal-component analysis to select a small sample of EOSs that characterize the 68% credible region of the distribution. This is done as follows:

1. construct a normalized set of variables defined by $\hat{x} := (x - \mu_x)/\sigma_x$, with $\mu_x$, $\sigma_x$ the mean and standard deviation for the variable $x$.
2. construct the 4 × 4 covariance matrix of these normalized variables.
3. calculate the eigenvalues $\lambda_i$ and eigenvectors $v_i$ of this covariance matrix, ordered by the magnitude on the eigenvalues.

The orthogonal vectors $v_i$ define the principal components of the distribution in the original 4D space, while the $\lambda_i$ characterize the variance of the distribution in each of these directions. In principle, one could generalize this analysis by considering additional uncorrelated variables to the four we have chosen, or even attempt to identify the optimal set of uncorrelated variables, but this is beyond the scope of this work.

Figure 5 shows on the top panel the components $v_i$ in the normalized coordinate system, while the bottom panel shows the posterior distribution in the $v_i$ coordinate system.

As seen in this figure, the distribution is primarily 3D, with a prominent triangular component within the plane spanned by the components $v_0$ and $v_1$. This behaviour of the distribution clearly explains a well-known aspect to anyone constructing agnostic models of EOSs, namely, that while it is reasonable to model the variation in EOSs in terms of stiffness, i.e., $\hat{R}_{1.4}$, this choice does not cover all of the possible space of parameters, which can be determined for instance in a principal-component analysis. Finally, we select the six golden EOSs from our ensemble by choosing EOSs that are near the extrema of the 68% credible region and one near the origin. With a standard principal-component analysis, these points would be given by $\pm\sqrt{\lambda_i}\,v_i$ (no summation), which we modify slightly to capture the triangular shape within the plane spanned by $v_0$ and $v_1$. In this plane, we use the

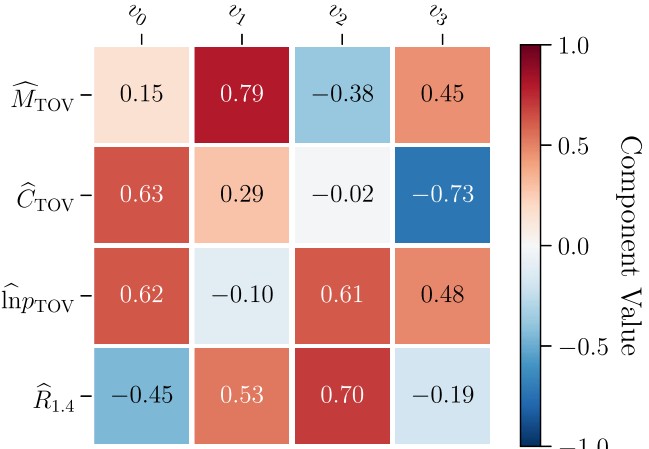

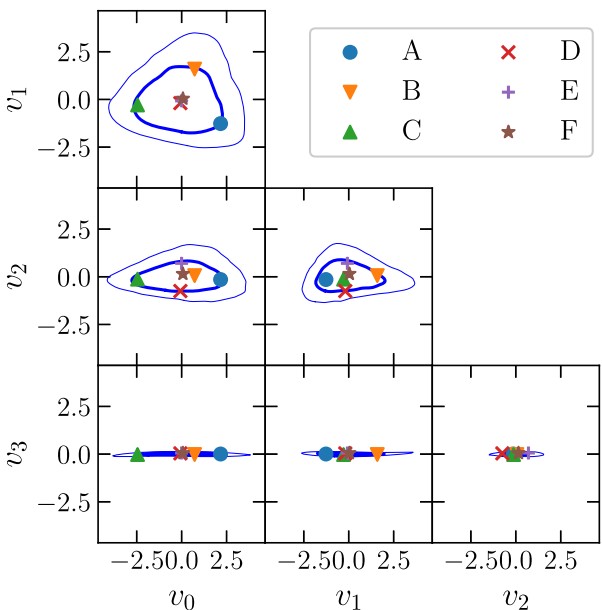

**Fig. 5 | Principal-Component Analysis.** *Left panel:* Components of the principal-component vectors $v_i$ in terms of the original (normalized) coordinates $(\hat{M}_{TOV}, \hat{C}_{TOV}, \hat{\ln} p_{TOV}, \hat{R}_{1.4})$. *Bottom panel:* Posterior distribution in the principle component analysis (PCA) coordinate system. Shown are 95% (thin lines) and 68% (thick lines) credible regions. Also shown are the six golden EOSs, with A–E lying on the 68% contour by construction, and F lying at the center. Source data for this figure are provided as a Source Data file.

directions that extremizes the 95% credible regions. Having identified the relevant points in the parameter space, we then select the golden EOSs corresponding to one of these six points by finding the 30 closest EOSs using the Euclidean metric in the full 4D space and selecting the one with the highest likelihood. (For a short discussion of the robustness of this procedure, see SI note 1.) We note that using the reduced 3D metric obtained by dropping the $v_3$ component leads to the same final golden EOSs. It is worth noting that using the corners of the 68% credibility contours for the golden EOS selection is a matter of choice in how we represent the underlying distribution. The variability of the simulations with the golden EOSs can be used as a proxy to approximate the 68%-credible regions that would be obtained if the full GP ensemble was used. In our analysis, this choice of 68% represents a compromise between capturing the extrema of the EOS distribution and assuring a sufficiently high posterior probability for the selected EOSs. In particular, choosing instead the 95% contour would represent the same distribution with a selection of EOSs coming from the tails of the EOS distribution, that is, EOSs that have significantly lower likelihood. This choice does not affect the overall results to the extent that our golden set characterises the features of the underlying distribution.

## Merger simulations and GW analysis

The initial data for our simulations is generated using the spectral-solver code `FUKA`[43], which allows us to produces both equal and unequal mass irrotational BNS initial configurations with an initial separation of approximately 45, km. `FUKA` employs the extended conformal thin-sandwich formulation of Einstein's field equations to solve for binaries under the quasi-circular orbit approximation. Residual eccentricities are minimized by incorporating estimates for orbital and radial infall velocities at 3.5PN order[43].

For the evolution, we employ the `Einstein-Toolkit`[44], which incorporates the fixed-mesh box-in-box refinement framework `Carpet`[45]. Specifically, we utilize six refinement levels, with the finest grid having a spacing of 295, m, and impose reflection symmetry across the orbital plane. This resolution enables us to explore a reasonable portion of the EOS and BNS parameter space while maintaining manageable computational costs. The computational domain extends to an outer boundary at ±1512, km, ensuring accurate gravitational wave extraction and appropriate boundary conditions.

For the spacetime evolution, we utilize `Antelope`[46], a solver for the constraint-damping formulation of the Z4 system[47]. The evolution of matter is handled using the `FIL` general-relativistic magnetohydrodynamic code[46], which employs fourth-order conservative finite-difference methods, ensuring high accuracy in hydrodynamic evolution even at the resolution adopted in this study. Additionally, `FIL` supports tabulated EOSs that depend on temperature and electron fraction and incorporates a neutrino transport scheme capable of modeling neutrino cooling and weak interactions[48]. To maintain our description as simple as reasonably possible, we have considered zero magnetic fields and neglected the radiative transfer of neutrinos; while we do not expect qualitative changes from either magnetic fields or neutrinos, it is reasonable to expect will play a quantitative role in establishing the long-ringdown slope.

As discussed in the main text, our agnostic EOS construction provides the cold (i.e., $T = 0$, where $T$ is the temperature) part of the EOSs, while the thermal part is added during the evolution to account for shock-heating effects during and after the merger[22]. More specifically, the total pressure is given by the sum of the cold EOS $p_c = p_c(\rho)$ and a thermal component $p_{th} = p_{th}(\rho, T) = \rho T$, where $\rho := m_B n$ is the rest-mass density and $m_B = 931.5$ MeV the atomic mass unit. Analogously, the total specific internal energy can be separated into a cold $\epsilon_c = \epsilon_c(\rho)$ and a thermal part $\epsilon_{th} = \epsilon_{th}(T)$. Here, $\epsilon_c(\rho) = e_c(\rho)/\rho - 1$ and $\epsilon_{th}(T) := T/(\Gamma_{th}-1)$, where $e_c(\rho)$ is the energy density of the cold EOS and $\Gamma_{th}$ is the thermal adiabatic index, which we choose to take the constant value

$\Gamma_{th} = 1.75$. As a result, the total pressure and total internal energy density are given by

$$p(\rho, T) = p_c(\rho) + \rho T, \qquad \epsilon(\rho, T) = \frac{e_c(\rho)}{\rho} - 1 + \frac{T}{\Gamma_{th} - 1}, \qquad (5)$$

where the cold contributions $e_c(\rho)$ and $p_c(\rho)$ are provided in tabulated form by the GP construction explained above. The entropy can then be expressed as

$$s(\rho, T) = \frac{1}{\Gamma_{th} - 1}\left(\frac{\bar{\epsilon}_{th}}{\rho^{\Gamma_{th}-1}}\right), \qquad (6)$$

where $\bar{\epsilon}_{th} := \max(\epsilon_{th}, s_{min})$, with some numeric lower bound for the entropy $s_{min} = 10^{-10}$. In summary, our construction realizes a model-independent parametrization for the density and temperature dependence of viable NS EOSs without any information on the particle composition. Furthermore, since we neglect neutrino emission and absorption, no composition dependence is present in our simulations. Future analyses with self-consistent temperature-dependent EOSs could additionally explore a possible dependence of the long-ringdown slope on composition.

For the GW analysis, we adopt the Newman-Penrose formalism, which relates the Weyl curvature scalar $\psi_4$ to the second time derivative of the polarization amplitudes of the GW strain $h_{+,\times}$, as described in ref. [49]

$$\ddot{h}_+ + i\ddot{h}_\times = \psi_4 := \sum_{\ell=2}^{\infty}\sum_{m=-\ell}^{m=\ell} \psi_4^{\ell,m} \, _{-2}Y_{\ell,m}, \qquad (7)$$

where $_sY_{\ell,m}(\theta, \phi)$ are spin-weighted spherical harmonics with a weight of $s = -2$. From our simulations, we extract the multipoles $\psi_4^{\ell,m}$ at a sampling rate of about 634 kHz from a spherical surface of approximately 574 km radius, centered at the origin of the computational domain. The extracted data is then extrapolated to the estimated luminosity distance of 40 Mpc, corresponding to the GW170817 event[50]. We fix the angular dependence of the spherical harmonics by adopting a viewing angle of $\theta = 15°$, as inferred from the jet orientation of GW170817[51], and set $\phi = 0°$ without loss of generality. Our analysis is restricted to the multipoles with $\ell \leq 4$, noting that the $\ell = |m| = 2$ modes dominate the signal. The relative difference in the maximum gravitational wave amplitude, $\sqrt{h_+^2 + h_\times^2}$, when comparing results including multipoles up to $\ell \leq 4$ and only $\ell = 2$, is less than 3%. All results are presented as functions of the retarded time $t - t_{mer}$, where $t_{mer}$ is defined as the time corresponding to the global maximum of the gravitational wave amplitude.

A key quantity in our analysis is the instantaneous GW frequency, $f_{GW}$, defined as

$$f_{GW} := \frac{1}{2\pi}\frac{d\phi}{dt}, \qquad \phi := \arctan\left(\frac{h_\times^{2,2}}{h_+^{2,2}}\right). \qquad (8)$$

The radiated power is given by the integral expression[49]

$$\dot{E}_{GW} = \frac{r^2}{16\pi}\sum_{\ell=2}^{\infty}\sum_{m=-\ell}^{\ell}\left|\int_{-\infty}^t dt' \psi_4^{\ell m}\right|^2, \qquad (9)$$

where the total emitted GW energy follows from another time integration $E_{GW}(t) := \int_{-\infty}^t dt' \dot{E}_{GW}(t')$. Similarly, the rate of radiated angular momentum is defined as[49]

$$\dot{J}_{GW} := \frac{r^2}{16\pi}\text{Im}\left\{\sum_{\ell=2}^{\infty}\sum_{m=-\ell}^{\ell} m\left(\int_{-\infty}^t dt' \psi_4^{\ell m}\right)\int_{-\infty}^t dt'\int_{-\infty}^{t'} dt'' \bar{\psi}_4^{\ell m}\right\}, \qquad (10)$$

where $r$ is the observer distance and where $\bar{\psi}_4$ is the complex conjugate of $\psi_4$ and the total emitted angular momentum follows again from another time integration $J_{GW}(t) := \int_{-\infty}^{t} dt' \dot{J}_{GW}(t')$.

In the main text we work with dimensionless energies $E_{GW}(t)/E_{GW}^{mer}$ and angular momenta $J_{GW}(t)/J_{GW}^{mer}$ obtained by normalizing with the respective values at merger time $E_{GW}^{mer} := E_{GW}(t_{mer})$ and $J_{GW}^{mer} := J_{GW}(t_{mer})$. When expressed in terms of strain components, and in full generality, the ratio of the radiated energy and angular-momentum rates $dE_{GW}/dJ_{GW}$ is similar to the instantaneous GW frequency

$$\frac{dE_{GW}}{dJ_{GW}} = \frac{\dot{E}_{GW}}{\dot{J}_{GW}} = \frac{\dot{h}_+^2 + \dot{h}_\times^2}{h_+ \dot{h}_\times - \dot{h}_+ h_\times} \,, \qquad f_{GW} = \frac{1}{2\pi} \frac{h_+ \dot{h}_\times - \dot{h}_+ h_\times}{h_+^2 + h_\times^2} \,. \quad (11)$$

For a simple system with an $\ell = 2$, $m = 2$ deformation, e.g., a compact rotating system with eccentric mass distribution like the toy model of ref. 10 and for which $h_+(t) \propto \cos(\phi(t))$ and $h_\times(t) \propto \sin(\phi(t))$ with GW phase $\phi(t)$, one obtains the identity $\dot{E}_{GW}/\dot{J}_{GW} = f_{GW}/(2\pi)$. Since in the long ringdown $f_{GW}(t) \simeq \mathrm{const.} =: f_{rd}$, expressions (11) explain why the radiated energy and angular momentum are linearly related.

Finally, we analyze the spectral features of the waveforms and compute the PSD of the signal as[10]

$$\tilde{h}^{\ell,m}(f) := \frac{1}{\sqrt{2}} \left( \left| \int dt \, e^{-2\pi i f t} h_+^{\ell,m}(t) \right|^2 + \left| \int dt \, e^{-2\pi i f t} h_\times^{\ell,m}(t) \right|^2 \right)^{1/2}, \quad (12)$$

where the time integration is performed over the interval $t - t_{mer} \in [0, 30]$ ms or up to the time at which a black hole is formed if the post-merger remnant collapses earlier. As done routinely (see, e.g., refs. 9–14), the dominant post-merger frequency $f_2$ is then determined by the global maximum of the PSD.

## Data availability

The data generated in this study are available from the authors upon request. Furthermore, the data in the Figures and Tables in the main text and Supplementary Information file are provided in the accompanying Source Data file. Source data are provided with this paper.

## Code availability

The initial data has been computed with the publicly available `FUKA` code[43]; since magnetic fields are ignored, the evolutions can be reproduced with the publicly available `WhiskyTHC` code[52] and the `Einstein Toolkit`[44]; the code for the analysis of the waveforms is `Kuibit`[53]; the custom codes for the construction of the EOSs and PCA are available upon request from the authors.

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

## Acknowledgements

We thank M. Cassing, M. Chabanov, E. Most, C. Musolino, H.H.-Y. Ng, D. Radice, and K. Topolski for discussions and comments during the development of this work. Partial funding comes from the Deutsche Forschungsgemeinschaft (DFG, German Research Foundation) project-ID 279384907–SFB 1245, the State of Hesse within the Research Cluster ELEMENTS (Project ID 500/10.006), by the ERC Advanced Grant "JET-SET: Launching, propagation and emission of relativistic jets from binary mergers and across mass scales" (Grant No. 884631). C.E. acknowledges support by the DFG through the CRC-TR 211 "Strong-interaction matter under extreme conditions"—project number 315477589—TRR 211. L.R. acknowledges the Walter Greiner Gesellschaft zur Förderung der physikalischen Grundlagenforschung e.V. through the Carl W. Fueck Laureatus Chair. The simulations were performed on the local ITP Supercomputing Clusters Iboga and Calea and on HPE Apollo HAWK at the High Performance Computing Center Stuttgart (HLRS) under the grant BNSMIC.

## Author contributions

The identification of the golden EOSs was performed by T.G. and A.K. The numerical simulations were performed by C.E. with input from L.R. C.E., T.G., A.K., and L.R. all contributed to the analysis of the simulation results and the writing of the manuscript.

## Funding

## Competing interests

The authors declare no competing interests.
