## [Transparent Peer Review file · Nature Communications]

Constraining the equation of state in neutron-star cores via the long-ringdown signal

Corresponding Author: Dr Christian Ecker

Version 0:

Reviewer comments:

Reviewer #1

(Remarks to the Author)

Report: Listening to the long ringdown: a novel way to pinpoint the equation of state in neutron-star cores

The authors propose a new method for extracting information about the high-density equation of state of baryonic matter by considering the long ringdown post-merger signal from a binary neutron star merger, in particular the ratios of energy and angular momentum losses. They consider a large set of EOSs generated using a Gaussian process approach and imposing several constraints, ranging from observational to ab-initio calculations. From this large set, the authors select what they call the golden six EOSs and perform a principal component analysis using as variables the normalized TOV mass, TOV compactness, TOV pressure and $R_{1.4}$. A 3D distribution was obtained and they selected approximately the 68% extremes in three of the vectors and one in the center. Using these EoS, they performed merging simulations for three different mass asymmetries and found a strong correlation between $dEGW/dJGW$ and $pTOV$ and $nTOV$, stronger than the corresponding correlation already known for f_2 . This allowed them to infer the EOS from the knowledge of the slope EGW/JGW and the frequency f_2 with a better accuracy than using f_2 alone. The possible determination of the behavior of high-density baryonic matter from the observation of NS is certainly one of the major problems in the field. It is therefore important to establish new and more precise approaches to achieve this goal. However, some points are not very clear and we would like to ask the authors for clarification.

Why have the authors restricted the choice of EoS to the 68% credible region of the distribution on the space of normalized $MTOV$, $pTOV$, $CTOV$, $R_{1.4}$, when the other plots identify the 95% credible regions ($p(n)$, TOV and long ringdown slope)?

Could it be that limiting the analysis to the 68% credible region is too restrictive?

The method used to choose the six golden EoS is clear, but it seems that the different properties spanned by this set of EoS is small: in figure 1 left panel the EoS between 2 and 3 ρ_0 are all very similar except for A , all the MR curves except one have a positive slope in the range 1-1.7 M_{sun} , all EoS except one predict a maximum masses within $2.17 \pm 0.05 M_{\text{sun}}$, all central densities except one are within $5.30 \pm 0.5 \rho_0$. The 1.4 M_{sun} radius predicted by 4 of the EoS differs by less than $\sim 350\text{m}$, a difference that may be of the order of the uncertainty introduced for using the same crust for all EOS. One observes that the MR curves only cross each other above $2M_{\text{sun}}$, and they are essentially parallel to the 95% contours. No NS has a compactness above 0.2 (five seem to lie between 0.185-0.198 and one ~ 0.17 . Note that the compactness in Table 1 is not correctly calculated). Could it be that choosing $RTOV$ as one of the variables would have spanned a larger domain (a large $R_{1.4}$ and a small $RTOV$). The variables $MTOV$, $CTOV$ and $pTOV$ seem to be too correlated. Two very frequently used EOS in SN and BNS simulations are DD2 and SFHo, both have a compactness just above 0.2. DD2 has a maximum mass of $2.42 M_{\text{sun}}$, $R_{1.4} = 13.3\text{km}$, $RTOV = 12.1\text{km}$ and TOV pressure = $0.432 \text{ GeV}/\text{fm}^3$, SFHo has a maximum mass of $2.07 M_{\text{sun}}$, $R_{1.4} = 11.9\text{km}$, $RTOV = 10.3 \text{ km}$ and TOV pressure $0.716 \text{ GeV}/\text{fm}^3$. Both are outside the domain spanned by the golden six. In order to introduce temperature effects, the authors have introduced an effective temperature contribution, in particular, using the adiabatic index 1.75. However, it is not clear how the composition is taken into account in the simulation. To test their approximations, the authors have also considered the evolution considering two tabulated temperature-dependent EOSs, DD2 and V-QCD. Could the authors include the result of these simulations in the corresponding panels of Fig. 3 and add the information to Table 1?

The authors say that they fitted Eq. 1 over the time window $t-t_{\text{mer}}$ in $[1,30]$ ms. Could the authors clarify what was done?

What values from the simulation were used to calculate the slope? It appears that Eq. 1 is fitted using the 18 points from Table 1. Are 18 points sufficient to fit a surface accurately in 3-dimensional space? In order to include information about f_2 in the inference shown in Figure 4, a similar fit was performed for this quantity. How efficient was this fit?

What would have been the result of the inference plotted in Fig 4 taking only dE/dJ information?

Figure 4 is one of the main results of this study. The figure shows the dataset ensemble with 95% credible intervals (CI), and

within the inference performed with three different slopes and corresponding f_2 frequencies obtained at 68% CI. Assessing the constraint capacity of Eq. 1 with different CI from the dataset ensemble can be misleading

Other comments

Is the definition of f_{GW} in line 142 correct? (pg 4 third line before the end)

Reviewer #2

(Remarks to the Author)

Reviewer #3

(Remarks to the Author)

To the kind attention of Nature Communication editors and of the authors of the manuscript "Listening to the long ringdown: a novel way to pinpoint the equation of state in neutron-star cores".

The paper deals with an interesting and timely topic. The connection between GWs and the EOS of nuclear matter is certainly a very interesting and broad subject. The relation between the evolution of the energy and angular momentum emission in GWs and the NS properties presented in the paper is valuable. However, I have several concerns about the paper in its present version. These deal with the novelty of the work and with the validity of the presented conclusions. Additionally, I have some remarks about the method and about the manuscript presentation. Below I detail my comments.

=====
Key results
=====

In this paper, the Authors present a novel way to constrain the EOS of dense matter through the analysis of the GW post-merger signal produced by a remnant that does not collapse to a BH within the first 10-20ms after merger, i.e. of a HMNS remnant.

By analyzing the outcome of merger simulations in Numerical Relativity done by using state-of-art codes in the field, they find a correlation between the ratio at which energy and angular momentum are radiated in GWs, and maximum pressure and density that characterize maximal TOV solutions obtained by a certain EOS. By performing mock measurements of post-merger signals, they find an improvement in the capability of setting constraints on the EOS at several times nuclear saturation density.

=====
Validity and Novelty
=====

I have a few concerns about the novelty and the validity of the paper.

* In lines 122-124 it is stated that the temporal behavior of E_{GW} and J_{GW} was never appreciated so far.

However, the behavior depicted in the top panel of figure 2 was already presented and commented in some other works, see e.g. figure 2 of [10.1103/PhysRevD.94.024023] or figure 5 of [10.1103/PhysRevD.102.024087].

In the first one, the linear relation between E_{GW} VS J_{GW} is evident, as well as a possible dependence on the EOS for the post-merger signal.

In the latter, it is also visible the fact that other important parameters of the binary (e.g. the NS spins) can have a similar effect on the E_{GW} VS J_{GW} evolution.

Refs. [30-31] are actually indicated as the first place where the longterm behavior of f_{GW} was discussed, but my understanding is that also the top panel of fig.2 was already presented there.

* The true novelty of this work seems to me the correlation represented by eq 1, where the slope and the EOS properties are directly connected.

However, empirical/analytical models of the post-merger GW signal, as for example the one presented in ref [32] or in [10.1088/0264-9381/33/8/085003] or in [10.1103/PhysRevD.100.043005], were also calibrated against the outcome of Numerical Relativity simulations.

So they likely include the correct evolution of E_{GW} and J_{GW} as a function of time, and thus of the ratio of their derivatives.

So, despite the fact that they do not contain an explicit relation between the value of the E_{GW} VS J_{GW} slope and the EOS-dependent parameters, it seems to me that an analysis perform with those post-merger models should implicitly include that information.

What do the authors think? In other words, in which sense should their approach be different from or better than a post-merger GW model that reproduces the same feature in inferring the EOS properties, as it was done for example in [10.1103/PhysRevD.105.104019] or [10.1103/PhysRevD.100.104029] or [10.1103/PhysRevLett.120.031102]?

* Concerning the mock measurement illustrated in lines 200-216 and in the "Performing the mock measurement" section in the SM, my understanding from the text is that the authors use a multivariate Gaussian distribution for f_2 and for the normalized dE/dJ true value, assigning in both cases a 8% uncertainty in a way which looks very arbitrary to me.

How realistic is it to assume the same uncertainty for f_2 (the peak of the GW PSD) and for the slope of J_{GW} VS E_{GW} (which must be determined from the GW signal, together with their values at merger, since the correlation was studied for the normalized ratio)?

Wouldn't it be more correct to simulate a certain signal, with a certain SNR, and extract from there all the relevant quantities to have more meaningful estimate of the uncertainties? Alternatively, how much does the result depend on the assumption that the slope of the normalized J_{GW} VS E_{GW} relation have the same uncertainty as the f_2 peak?

=====
Significance
=====

The relationship between the normalized slope and the EOS-dependent NS parameters is certainly interesting. It has the potential to be useful in adding a new piece of information in future GW analysis. However, the significance that I see is possibly more limited than the one presented by the authors.

* Some of the statements that are supposed to stress its potential significance seem to me a bit overrated. For example, I do not agree with lines 185-187. The density related to low mass NS refer to the inspiral phase. However, after merger, the density inside the HMNS increases and is probably a significant fraction of n_{TOV} . So, it is not so surprising to me that a relation deduced from the post-merger signal set constraints on the high density part of the EOS. Moreover, according to lines 213-215, the improvement due to the inclusion of the novel correlation is "considerable", with respect to the case in which only f_2 was considered. Honestly, looking at figure 4, the improvement looks rather limited to me (~20% level). Moreover, this was obtained by considering a rather small and equal errors on f_2 and on the normalized E_{GW} VS J_{GW} slope. How would perform an analysis that uses only the normalized slope, i.e. without f_2 ? How do the results change with respect to the error on the slope?

* In addition to providing further constraints on the functional relations $P=P(n)$ and $M=M(R)$, it seems to me that, in the way it was presented, this relation has little impact on our understanding on the physics that regulates the EOS. Stated differently, to which extend this relation has the potential of distinguishing between different nuclear models, e.g. models that feature a strong phase transition, a smooth phase transition, or no phase transition to quark matter? Such a potential could be very interesting, while a further generic constraint would be very similar to the outcome of several previous works on EOS-constraints from the post-merger signal, maybe based on models that reproduced internally the same relation stressed in this work.

=====
Data and methodology, with Suggested improvements
=====

The method is overall fine and the analysis method looks correct. However, also in this case, some potential flaws or weaknesses are present.

* At line 328-329, the authors write about an "intermediate resolution".

What does intermediate mean here? Between what?

I completely understand that a large parameter space does not allow high resolution simulations, but ~300m looks rather coarse to me.

Did the authors try to estimate the potential impact of resolution on their results?

* In the conclusions, it is written that the correlation is robust against:

- the chirp mass;
 - unequal mass binaries;
 - temperature dependent EOSs;
 - different values of the thermal index;
- supported by the paragraph "On the robustness of the correlation" in the SM. It seems to me that this is not really supported for all variables.

In particular:

- The tests on M_{chirp} indicate that the impact of M_{chirp} is small.

Moreover, since M_{chirp} will be determined with high accuracy for GW signals with high SNR, even a residual small dependence is not a big issue. So, this is fine.

- Moving to the unequal mass cases, I do not understand why the $q=0.7$ and $q=0.85$ cases are separately discussed in lines 395-410 and why in the conclusions it is written that the "novel correlation" applies also to them. To my understanding, also such cases were included in the determination of the beta coefficients of eq (1) and the were present in fig 3. So, why do the authors check the robustness of the correlation against cases used to compute the correlation itself?

It seems to me that in the SM the authors simply show more details about the unequal mass mergers.

- The thermal index analysis is analogue to the chirp mass analysis, so it is ok.

- The conclusion about the temperature dependent EOS is not correct. The only think that the SM shows is that in simulations employing temperature-dependent EOSs the evolution of E_{GW} VS J_{GW} is qualitatively similar to the one obtained with the thermal index EOS (which was already implied by the simulations showed in [31], for example). But this does not imply the quantitative (or even qualitative) validity of the correlation found by employing EOSs characterized by the thermal index also for the other EOSs.

How do the correlation presented in eq (1) and table 2 applies to the results obtained with the HS-DD2 and the V-QCD EOS?

Such a test could be beneficial.

* Finally, at line 175, if I understand correctly, a correlation like eq 1 is also applied to f_2 to test the impact of a single f_2 VS a joint (f_2 +slope) analysis. Is it correct? If so, why such a correlation is used also for f_2 ?

Would it be possible that other functional relations work better for f_2 ?

Are there other correlations involving f_2 in the literature?

Moreover, I do not understand the "respectively" in the text. What is the second term?

=====
Clarity and context
=====

The text is overall well written.

However, the flow is not always optimal. And some sentences are unclear or partially incorrect.

* At line 47, it is stated that the remnant of GW170817 collapsed to a BH ~1 second after merger. While I think there is a general consensus that GW170817 did not promptly collapsed to a BH, the fact that it survived for ~1 sec is not so well established. The analysis reported in [5] is valuable, but requires a certain number of assumptions. Other analysis seem to point to shorter lifetime (for example, a long lived remnant could have produced a blue kilonova only or a much brighter MHD-powered kilonova,

see e.g. [10.1093/mnras/stx1987] or [10.1093/mnras/stu802].
Due to the large number of uncertainties in our understanding, I would be more cautious or generic about this point.

* The first sentence after line 200 ("Having pointed out...") is unclear to me. It seems that there is a missing piece implied by "between".

* I am confused by the line of reasoning contained in lines 149-176.

A) In 149-152 it is written that for a signal with a high enough SNR the slope between E_{GW} and J_{GW} can be measured accurately. But more details about the way to extract such a slope, as well as the uncertainties on it, are not provided.

B) In 156-159 it is written that f_{rd} and the slope between E_{GW} and J_{GW} have the same value. But this is not true: as it is written at line 160 and one can verify also later in the text, the two quantities are conceptually analogue, but different. For example, according to the simulation results, f_{rd} is much closer to f_2 than the normalized slope.

C) The statement that the slope is much robust to fit than f_{GW} (lines 160-161) is not obvious: this depends on the uncertainties that we can associate to the quantities we have to fit. If E_{GW} and J_{GW} have significantly larger uncertainties than f_{GW} , this cannot be the case. Moreover, it is unclear where this is discussed more in the SM.

D) How are the "extrapolated slopes" of figure 2 determined?

E) How are the Pearson-correlation coefficient quoted in lines 171-176 computed? It is unclear from which analysis they are deduced.

Overall, in addition to fixing point B), I think more details are necessary to support the other points.

Reviewer #4

(Remarks to the Author)

The authors propose a new method for connecting measurable quantities from gravitational-wave observations of long post-merger parts of binary-neutron-star mergers and the equation of state of high-density matter, which is not accessible through experiments or observations of stable neutron stars. With a statistical analysis, they also show how the new method improves on previous methods and highlight its applicability to future observations. In detailing the many advantages of the new method, the authors also provide plenty of convincing comments on possible objections a reader might have to its generality and applicability.

The codes used in this work have been thoroughly tested and there is no reason for doubting the results of the simulations, but I checked with the data of our own simulations and found supporting evidence.

This is an important work, which will very likely allow more accurate identification of the properties of the equation of state in future gravitational-wave measurements. And, thus, it is a significant advance in the field.

The presentation of the background, idea, setup and results is very good: clear, detailed, and accessible to a range of physicists with different specializations.

Below are some questions and comments the authors may consider.

1) On lines 107, ref. [24] is cited for $\gamma_{\text{thermal}} = 1.75$. However, Figura et al. PRD102 (which is ref. [24]) state that: "we have concluded that a value of $\gamma_{\text{thermal}} \approx 1.7$ best approximates the complete, finite-temperature EOS in binary NS simulations. This value is similar to the standard one employed in numerical simulations so far (i.e., $\gamma_{\text{thermal}} = 1.75-1.80$), but also importantly lower."

It is customary to use $\gamma_{\text{thermal}} = 1.75$ in BNS simulations and there is no problem with that, but I recommend the authors to cite a more

appropriate reference than [24].

2) In the sentence on lines 184-7, can the second statement ("and we find it quite remarkable...") also be deduced directly from Fig. 3? If so, the authors may clarify how.

Finally, a very small lexical comment:

On line 163, maybe "transparent" is not the right term for describing the lines in paler colors.

Version 2:

Reviewer comments:

Reviewer #1

(Remarks to the Author)

The authors have addressed all of our comments and have supplemented the text where appropriate. However, we still believe that the same statements should be made more carefully, taking into account the conditions under which the study was carried out, in particular the fact that the analysis was carried out considering EOS within the 68% CI.

1. The authors write "In principle, one could generalize this analysis by considering additional uncorrelated variables to the four we have chosen, or even attempt to identify the optimal set of uncorrelated variables, but this is beyond the scope of this work."

The question is precisely whether the chosen set of uncorrelated variables is the optimal one, and whether a different set would not lead to a "golden" EOS with different behavior. Even if this is beyond the scope of the present study, the need to identify the optimal set of uncorrelated variables should be made more explicit. Besides the authors' option of considering a 68% CI does not leave much room for extra uncertainties.

2. The authors write "The Referee correctly points that different theoretical treatments may lead to differences in R1.4. This is, however, not a valid point of criticism towards our work as our EOSs arise from a model-independent prior which has been empirically constrained to this range. Deviation of this range would necessarily lead to being in tension with the LIGO tidal-deformability measurement and/or the existence of two-solar-mass stars."

We agree with the statement that the EoS should cover the region defined by NICER and LIGO in M-R space, but it has been shown that a non-unified crustal EOS can introduce uncertainties in the radius estimate, and therefore the model-independent EOS determined by the observational constraints is affected by the model-dependent crust (BPS crust), see e.g. Davis et al. 2406.14906 [astro-ph.HE] and references therein

3. The authors write: "In particular, choosing instead the 95% contour would result in a selection of more exotic EOSs, that is, EOSs that have significantly lower posterior likelihood, without impacting the overall results of the our analysis." It is not clear how the authors can say "without affecting our analysis" as this has not been tested. It is also not clear what is meant by exotic. According to the authors' interpretation, it seems to be equivalent to being outside the 68% CI, and therefore SFHo is an exotic EOS. However, one would not expect the samples to be analyzed and interpreted as "exotic cases" after the CI has been defined.

According to the methodology presented, SFHo is an exotic EOS, although by construction it satisfies nuclear properties - properties not imposed in the present approach - in addition to observational constraints. One would not say that it is an exotic EOS. The low-density constraints imposed in the present study are those introduced by the beta-equilibrium EOS constructed in Hebeler et al. 2013 from the neutron matter EOS determined in a chEFT approach and an effective description of symmetric matter. The band defined by the soft and hard EoS in the 0.5-1.1 n_0 was considered to define a 90% CI. It is difficult to accept that there is more information on nuclear matter at and below the saturation density with the new method introduced in the present work than the one used to determine SFHo. Again, the choice of considering only 68% CI is quite strict to allow for possible uncertainties in the low density EOS.

The new method proposed to infer the behavior of high-density baryonic matter from the knowledge of the long post-merge is certainly important, and as the authors point out, this is a preliminary study that can be improved in a number of ways in the future, including those mentioned above.

Reviewer #2

(Remarks to the Author)

Reviewer #3

(Remarks to the Author)

To the kind attention of the editors and of the authors.

I read the authors' answers to my points, as well as of the other referees.

First of all, I would like to thank the authors for taking into account my concerns and in answering my questions.

In most of the cases, the answers were satisfactory and the changes have, in my opinion, improved the readability of the manuscript.

Concerning the degree of novelty, I think that the present version better represents the content development. The fact that E_{GW} and J_{GW} have a linear relation is not the true novelty of the work, while I agree with the authors that the relation with the EOS is the real novelty of the work.

I have a few further (minor) comments, which the authors can find below.

A) Concerning my previous comment:

> Referee: Refs. [30-31] are actually indicated as the first place where the
> long-term behavior of fGW was discussed, but my understanding is that also
> the top panel of fig.2 was already presented there.

> Our response: Unfortunately, we cannot follow Referee's comment. Our
> refs. [30-31] are not related to the long-term behaviour of fGW.

In the first manuscript, refs. 30 and 31 were:

[30] Bernuzzi, S., Dietrich, T., Nagar, A.: Modeling the Complete Gravitational Wave Spectrum of Neutron Star Mergers. Phys. Rev. Lett. 115(9), 091101 (2015) <https://doi.org/10.1103/PhysRevLett.115.091101> arXiv:1504.01764 [gr-qc]

[31] Bernuzzi, S., Radice, D., Ott, C.D., Roberts, L.F., Moesta, P., Galeazzi, F.: How loud are neutron star mergers? Phys. Rev. D 94(2), 024023 (2016) <https://doi.org/10.1103/PhysRevD.94.024023> arXiv:1512.06397 [gr-qc]

The second one is now reference [28], while reference [30] has now disappeared. My point was simply that these references were already present, but the present version improves upon the original one with respect to a better location in the text and a more appropriate usage/acknowledgment of previous findings. The present version is thus fine with me.

B) At line 56 the acronym PSD is introduced. I would define it as: "power spectral density" rather than "spectral power density".

C) I suggest the authors to read again the caption of figure 7. I cannot see the dashed lines representing f_2 . Moreover, I do not understand in the final sentence "...where and shaded...".

D) I thank the authors for providing more runs with different resolutions. I still think that $\Delta x \sim 300\text{m}$ is a rather coarse resolution for a BNS merger, but I agree that it is probably enough to extract the dominant GW frequencies with sufficient accuracy. Resolution is crucial, for example, for the remnant lifetime or the ejecta properties, but this is not the point of the paper. In lines 414-417, the authors seem to be surprised by this outcome. However, recent resolution studies (e.g. Zappa et al, MNRAS, Volume 520, Issue 1, 2023) confirmed that. Additionally, if possible, at line 360 I would remove "intermediate".

E) I checked the references and I have a few remarks.

* At line 107, references [12] and [19-24] should support the fact that the temperature in BNS mergers reaches tens of MeV. But all these references, with the exception of 23 and 24, used an hybrid EOS, in which thermal effects are not accurate or consistent.

And even in ref 23 and 24, the focus is on hadron-quark phase transition, i.e. the case not explored in the paper.

I suggest the authors to search for works in which the temperature and the thermodynamics conditions in BNS merger are analyzed in a more systematic way, using finite-T EOSs and possibly including also works in which no hadron-quark phase transition is present. While I would avoid references to work that uses hybrid EOS approaches.

* When commenting about the $l=2, m=1$ instabilities, the authors wrote that such mode could be "dominant ... at later times for highly asymmetric binaries". However, studies like Radice et al 2016, PhRvD, 94, 064011, suggest that also symmetric BNS can produce it. Could the authors comment on that?

F) In table 1 of the SM, f_2 , f_{rd} and dE_{GW}/dJ_{GW} are provided as "EOS properties". How were they computed? This is presently unclear from the text.

Version 3:

Reviewer comments:

Reviewer #1

(Remarks to the Author)

We thank the authors for considering all our comments and clarifying the manuscript. We recommend the present study for publication in Nature Communications since it proposes a new method of extracting the high-density baryonic matter EOS from the observation of NS mergers, in particular, from the post-merger GW signals. This is a region of the QCD phase diagram not reachable in the lab, one of the major problems in the field.

Reviewer #2

(Remarks to the Author)

Reviewer #3

(Remarks to the Author)

To the kind attention of the editors and of the authors.

I thank the authors for taking into account my comments and for answering my questions.

However, I am not fully convinced by points C, E and F. Below they can find my further answers.

> Referee: C) I suggest the authors to read again the caption of figure 7. I cannot see the dashed lines representing f_2 . Moreover, I do not understand in the final sentence "...where and shaded...".
> Our response: We thank the Referee for pointing out what is actually a mistake as dashed lines were used in a previous previous version of the figure. We corrected the sentence accordingly and modified the wording in the part that refers to the shaded regions for the error estimate of f_2 :
> In the right panel we mark with solid lines the dominant post-merger frequency f_2 where and shaded areas indicate a 8% relative error estimate.

I thank the authors for amending the figure and the capture.
I still do not understand the meaning of "...WHERE AND..." in the final sentence. Could they please rephrase?

> Referee: E) I checked the references and I have a few remarks. At line > 107, references [12] and [19-24] should support the fact that the temperature in BNS mergers reaches tens of MeV. But all these references, with the exception of 23 and 24, used an hybrid EOS, in which thermal effects are not accurate or consistent. And even in ref 23 and 24, the focus is on hadron-quark phase transition, i.e. the case not explored in the paper. I suggest the authors to search for works in which the temperature and the

> thermodynamics conditions in BNS merger are analyzed in a more systematic way, using finite-T EOSs and possibly including also works in which no hadron-quark phase transition is present. While I would avoid references to work that uses hybrid EOS approaches.
> Our response: While we appreciate the Referee's suggestion about making more accurate citations to simulations using realistic and temperature dependent EOSs, we feel the present references are appropriate given the very qualitative nature of the sentence, which refers to "tens of MeV" and not to a precise range of temperature. [We incidentally note that a very old paper measured temperatures of the order of tens of MeV despite using a very simplified $\Gamma = 2$ law (0804.0594)]

I am sorry for stressing again this point, but I do not agree with the answer. In my comment, I did not write that the estimate or the results of the present references were wrong or not valuable.

I simply wrote that the statement that the temperature in BNS mergers reaches tens of MeV is now supported by works and simulations in which a dedicated analysis is provided and in which the temperature is consistently taken into account.

To me, these seem to be the most appropriate references to that sentence.

If the authors want to keep in practice the present set of references, I am fine (most of these references are anyway used elsewhere in the manuscript), but I would at least add (or replace) one reference citing a paper in which:

- investigation of the thermodynamical conditions was done in a systematic way;
- finite-T, composition dependent (and, for consistency with the present work, possibly hadronic) EOSs were employed.

I would like to notice that in ref [12] (De Pietri et al PRL 2018), the word "temperature" does not appear at all. I am even doubtful if this reference is appropriate as a description of the "general merger dynamics" (as reported a few lines after), since it focuses on a possible GW emission happening on timescales much longer than the ones explored here.

Such a GW-emission is rather controversial and could be due to low resolution artifacts:

as the authors admitted, a resolution of $\Delta x \sim 300\text{m}$ is not appropriate to model the post-merger phase, and for the precise modeling of such long-term behavior magnetic field, finite-T, and composition effects are not negligible.

> Referee: F) In table 1 of the SM, f_2 , f_{rd} and $dEGW/dJGW$ are provided as "EOS properties". How were they computed? This is presently unclear from the text.

> Our response: Our procedure for computing $dEGW/dJGW$ is explained in detail on pp. 17-18 of the manuscript. We apply an identical procedure to determine f_{rd} , which we mention now explicitly for clarity on line 480:

> We apply an analogous procedure to determine f_{rd} .

> In addition, we mention now also that the values of f_2 listed in Table 1

> correspond to the global maxima of Eq. (12) on line 443:

> As done routinely (see, e.g., [9-14]), the dominant post-merger frequency f_2 is then determined by the global maximum of the PSD.

I thank the authors for better explaining the way in which the quantities appearing in table 1 were computed. This is helpful.

But I think I did not express my original point in a clear way. At the moment, f_2 , f_{rd} and $dEGW/dJGW$ are presented as "EOS and NS properties" (see bold text of the caption and the text after it: "For each EOS, we list..."), at the same level as the maximum NS mass or maximum pressure.

But all the quantities coming after the "q" column refer to (and depend on) BNS system modeled using a certain EOS. So, I find the caption misleading. Could the authors improve on this?

Version 4:

Reviewer comments:

Reviewer #3

(Remarks to the Author)

I thank the authors for further improving the manuscript based on my suggestions. The present version of the manuscript is fine for me.

Reviewers # 1 and # 2 (Remarks to the Author):

Referee: Report: Listening to the long ringdown: a novel way to pinpoint the equation of state in neutron-star cores The authors propose a new method for extracting information about the high-density equation of state of baryonic matter by considering the long ringdown post-merger signal from a binary neutron star merger, in particular the ratios of energy and angular momentum losses.

They consider a large set of EOSs generated using a Gaussian process approach and imposing several constraints, ranging from observational to ab-initio calculations. From this large set, the authors select what they call the golden six EOSs and perform a principal component analysis using as variables the normalized TOV mass, TOV compactness, TOV pressure and $R_{1.4}$.

A 3D distribution was obtained and they selected approximately the 68% extremes in three of the vectors and one in the center. Using these EoS, they performed merging simulations for three different mass asymmetries and found a strong correlation between dE_{GW}/dJ_{GW} and $pTOV$ and $nTOV$, stronger than the corresponding correlation already known for f_2 .

This allowed them to infer the EOS from the knowledge of the slope E_{GW}/J_{GW} and the frequency f_2 with a better accuracy than using f_2 alone. The possible determination of the behavior of high-density baryonic matter from the observation of NS is certainly one of the major problems in the field. It is therefore important to establish new and more precise approaches to achieve this goal. However, some points are not very clear and we would like to ask the authors for clarification.

Our response: We thank the Referee for a concise and accurate description of our work as well as noting the importance of the research question.

Referee: Why have the authors restricted the choice of EoS to the 68% credible region of the distribution on the space of normalized MTOV, $pTOV$, $CTOV$, $R_{1.4}$, when the other plots identify the 95% credible regions ($p(n)$, TOV and long ringdown slope)? Could it be that limiting the analysis to the 68% credible region is too restrictive?

Our response: We thank the Referee for this interesting question. Restricting to 68% or 95% credible intervals is a matter of choice but the

qualitative features of our analysis will not depend on this choice. This is because, as shown with thin blue lines in Fig. 5, the 95% contours of the EOS distributions have a shape that is very similar to the 68% contours, but the corresponding EOSs would be more exotic in the sense that they have significantly lower posterior likelihoods and therefore represent rather extreme examples of EOSs. So, our choice of using the corners of the 68% (one standard deviation) EOS contours and not the 95% to identify the golden EOSs is motivated by the need of capturing the extrema of the central region of the EOS distribution where the posterior likelihood is still reasonably high. In this sense, identifying golden EOSs on the 68% contour represents a compromise between capturing the extrema of the distribution while maintaining a sufficiently high likelihood of the EOSs.

Following the Referee’s comment, we have now added a comment in the main text of the revised version of the manuscript that clarifies this point [the new text is indicated in boldface blue in the revised manuscript]:

We have chosen this region so that our sample characterises the distribution where most of the posterior weight is; a different choice of, e.g., 95% would consist of EOSs that are already in tension with observations and would not necessarily characterise the distribution as faithfully as the sample would be sensitive to the tails of the distribution.

We have also added the following remark to the Methods section:

It is worth noting that using the corners of the 68% credibility contours for the golden EOS selection is a matter of choice. In our analysis, this choice represents a compromise between capturing the extrema of the EOS distribution and assuring a sufficiently high posterior probability for the selected EOSs. In particular, choosing instead the 95% contour would result in a selection of more exotic EOSs, that is, EOSs that have significantly lower posterior likelihood, without impacting the overall results of the our analysis.

Referee: The method used to choose the six golden EoS is clear, but it seems that the different properties spanned by this set of EoS is small: in figure 1 left panel the EoS between 2 and 3 ρ_0 are all very similar except for A, all the MR curves except one have a positive slope in the range 1-1.7 M_\odot , all EoS except one predict a maximum masses within $2.17 \pm 0.05 M_\odot$,

all central densities except one are within $5.30 \pm 0.5 \rho_0$. The $1.4 M_\odot$ radius predicted by 4 of the EoS differs by less than 350m , a difference that may be of the order of the uncertainty introduced for using the same crust for all EOS. One observes that the MR curves only cross each other above $2M_\odot$, and they are essentially parallel to the 95% contours.

Our response: As we mentioned earlier, the similarity of the EOSs is an important feature of our analysis as the spread of EOSs reflects our current uncertainty in the EOS. In particular the EOS below $2\rho_0$ is well constrained because of the combination of $2M_\odot$ and radius (and tidal deformability) measurements. The Referee correctly points that different theoretical treatments may lead to differences in $R_{1.4}$. This is, however, not a valid point of criticism towards our work as our EOSs arise from a model-independent prior which has been empirically constrained to this range. Deviation of this range would necessarily lead to being in tension with the LIGO tidal-deformability measurement and/or the existence of two-solar-mass stars. Hence, even if we were to change the description of the crust EOS, the bound on the $R_{1.4}$ would remain unchanged.

The differences among the different EOSs are small and this is indeed why it is generally so difficult to distinguish them in practice. Our approach in terms of PCA, however, provides a well-defined, mathematically sound procedure to select among closely resembling EOSs.

Referee: No NS has a compactness above 0.2 (five seem to lie between 0.185-0.198 and one 0.17. Note that the compactness in Table 1 is not correctly calculated).

Our response: Unfortunately, we are not able to follow the Referee’s comment and we have checked that the values reported in Table 1 are correct (please note that we adopt geometric units so that the compactness is dimensionless). However, let us clarify again that our golden EOSs are not selected “by hand” but are the result of the PCA applied a statistically large sample of EOSs. Furthermore, all of the golden EOSs have maximum compactnesses \mathcal{C}_{TOV} that are above 0.27, as reported in Table 1.

In order to be explicit about the units of the compactness we have added the word “dimensionless” to the following sentence:

The significant breadth of EOSs in the ensemble reflects the current level of uncertainty in the determination of the EOS. Because it is computationally

prohibitively expensive to scan a large number of EOSs, we reduce the full ensemble to a smaller sample of six “golden” EOSs that maximizes the variation in the following four NS parameters: the maximum (TOV) mass of an isolated, nonrotating NS M_{TOV} , its dimensionless compactness $C_{\text{TOV}} := M_{\text{TOV}}/R_{\text{TOV}}$, where R_{TOV} is the corresponding radius, the central pressure P_{TOV} , and the radius of a typical $1.4 M_{\odot}$ NS $R_{1.4}$.

Referee: Could it be that choosing RTOV as one of the variables would have spanned a larger domain (a large $R_{1.4}$ and a small RTOV). The variables MTOV, CTOV and pTOV seem to be too correlated.

Our response: Indeed, M_{TOV} , C_{TOV} , and p_{TOV} are strongly correlated, which is why we have decided to break this degeneracy by introducing in addition the quantity $R_{1.4}$. Furthermore, already when considering M_{TOV} and C_{TOV} as variables, including R_{TOV} would not improve our analysis since it is trivially correlated to M_{TOV} and C_{TOV} by $R_{\text{TOV}} = M_{\text{TOV}}/C_{\text{TOV}}$. That said, it may be interesting to generalize our analysis by introducing additional “uncorrelated” variables, or even attempt to identifying the “best” set of uncorrelated variables. Clearly, this goes beyond the scope of our work but is worth investigating in a separate analysis.

Following the Referee’s comment, we have now added a comment in the Methods sections of the revised version of the manuscript suggesting this extension of our analysis:

In principle, one could generalize this analysis by considering additional uncorrelated variables to the four we have chosen, or even attempt to identify the optimal set of uncorrelated variables, but this is beyond the scope of this work.

Referee: Two very frequently used EOS in SN and BNS simulations are DD2 and SFHo, both have a compactness just above 0.2. DD2 has a maximum mass of $2.42 M_{\odot}$, $R_{1.4}=13.3\text{km}$, $R_{\text{TOV}}=12.1\text{km}$ and TOV pressure= $0.432 \text{ GeV}/\text{fm}^3$, SFHo has a maximum mass of $2.07 M_{\odot}$, $R_{1.4}=11.9\text{km}$, $R_{\text{TOV}}=10.3 \text{ km}$ and TOV pressure $0.716\text{GeV}/\text{fm}^3$. Both are outside the domain spanned by the golden six.

Our response: Indeed, we have analyzed the HS-DD2 model in our Methods section where we discuss the robustness of our approach. We should

note that the HS-DD2 model does fall outside the domain spanned by the golden six and this is because it is in severe tension with the tidal deformability constraint deduced from the GW170817 event, which is by construction satisfied by our golden ensemble. The SFHo EOS is a viable model but is outside the 68% credible region of EOSs considered. (One can see this by comparing to Fig. 1 that the TOV pressure of the SFHo EOS lies at the upper boundary of the 95% credible region.) As we remarked above, our selection of the 68% credible region represents a compromise and the fact that some widely used EOSs, with much smaller likelihoods, are not within the span of our EOSs is an important feature of our method.

Referee: In order to introduce temperature effects, the authors have introduced an effective temperature contribution, in particular, using the adiabatic index 1.75. However, it is not clear how the composition is taken into account in the simulation.

Our response: We note that the composition dependence is relevant only in binary simulations in which the physics of neutrino emission and absorption is properly accounted for. However, this is not the case of our simulations and so such dependence cannot be explored in our model-independent approach for constructing the EOS, which does not include any information about the particle composition of the resulting EOSs.

Following the Referee’s comment, we have now added a comment in the Methods section of the revised version of the manuscript that clarifies this point:

In summary, our construction realizes a model-independent parametrization for the density and temperature dependence of viable NS EOSs without any information on the particle composition. Furthermore, since we neglect neutrino emission and absorption, no composition dependence is present in our simulations. Future analyses with self-consistent temperature-dependent EOSs could additionally explore a possible dependence of the long-ringdown slope on composition.

Referee: To test their approximations, the authors have also considered the evolution considering two tabulated temperature-dependent EOSs, DD2 and V-QCD. Could the authors include the result of these simulations in the corresponding panels of Fig. 3 and add the information to Table 1?

Our response: We thank the Referee for this useful suggestion. We have now added the corresponding information of our DD2 and VQCD simulations to Fig. 3 and Table 1. We have added the following sentence to the caption of Fig. 3:

Also shown are data points for two microscopic EOS models, one of which (HS-DD2) is disfavored by astrophysical data.

Referee: The authors say that they fitted Eq. 1 over the time window $t - t_{mer}$ in [1,30] ms. Could the authors clarify what was done? What values from the simulation were used to calculate the slope?

Our response: From the simulations one obtains the gravitational wave strain from which the emitted energy and angular momentum can be extracted using Eqs. (9)–(10). We have added details of the fit to the supplemental material:

Our final discussion on the methods employed in our analysis is focussed on the accuracy of the slope extraction from the numerical simulations. This operation requires first to identify the optimal time-range for the linear least-squares fit of $E_{GW}(J_{GW}(t))$, whose starting and final times are determined by minimizing the variance of the linear fit. More specifically, we first compute the slope and its variance for a number of different starting times $t_{in} - t_{mer} \in [1 - 10]$ ms using a fixed value for the final time $t_{fin} - t_{mer} = 15$ ms. In this way, we found that a starting time of $t_{in} - t_{mer} = 1$ ms results in an approximate variance of ± 0.1 for the slope. Larger values for the starting time, e.g. 5 ms or 10 ms, result in significantly larger variances of ± 0.7 and ± 3.1 , respectively. Next, we compute the slope and its variance for various values of the final time $t_{fin} - t_{mer} \in [2 - 25]$ ms while keeping the starting time fixed at $t_{in} - t_{mer} = 1$ ms. In this way, we found that the variance saturates at $t_{fin} - t_{mer} \approx 15$ ms to values similar to those obtained by varying the starting time. Increasing the final time of the fit does not lead to further improvement of the fit quality and this is because it becomes increasingly difficult to accurately compute the small changes in E_{GW} and J_{GW} at times beyond $t - t_{mer} \gtrsim 15$ ms, where the amplitude of the GW signal becomes very small.

Referee: It appears that Eq. 1 is fitted using the 18 points from Table 1. Are 18 points sufficient to fit a surface accurately in 3-dimensional space?

In order to include information about f_2 in the inference shown in Figure 4, a similar fit was performed for this quantity. How efficient was this fit?

Our response: We have verified that the intervals shown in Fig. 3 do not change appreciably if we remove the data for the $q = 0.85$ case, giving us confidence that the fit to the full 18 data points is robust. Regarding the fit of f_2 , the referee is correct that a similar fit was performed for this quantity. The quality of the fit to the f_2 data was of similar quality as the fit to the slope shown in the manuscript. (See also our response to a question by Referee 3 below.)

Since a question about this fit was also asked by Referee #3, we have added the following clarifying sentence to the text

For this analysis, we use a second bilinear model to fit the f_2 data, whose quality is found to be as good as the above fit to $d\hat{E}_{\text{GW}}/d\hat{J}_{\text{GW}}$.

Referee: What would have been the result of the inference plotted in Fig 4 taking only dE/dJ information?

Our response: The answer to the Referee's very interesting question is shown in the attached Fig. 1, whose top panels reproduce Fig. 4 from the manuscript, while the bottom panels show the credible intervals when taking into account *only* the $dE_{\text{GW}}/dJ_{\text{GW}}$ information. It should be noted that the credible intervals in this case are very similar (but different) from those obtained when using the f_2 information only, which are shown with dashed lines in the top panels of Fig. 1. However, taking into account information from *both* $dE_{\text{GW}}/dJ_{\text{GW}}$ and f_2 leads to noticeably smaller credible intervals.

Because this is one of the main messages of our manuscript, we have stressed this further in the revised version with the following text

We should note that similarly large credible intervals appear if we were to consider information on the slope only. Hence, Fig. 4 clearly highlights how the combination of information on the slope and on the f_2 frequency yields an increased accuracy in the properties of the EOS.

Referee: Figure 4 is one of the main results of this study. The figure shows the dataset ensemble with 95% credible intervals (CI), and within the inference performed with three different slopes and corresponding f_2 frequencies obtained at 68% CI. Assessing the constraint capacity of Eq. (1) with different CI from the dataset ensemble can be misleading

Figure 1: The top panels reproduce Fig. 4 from the manuscript, while the bottom panels show the credible intervals when taking into account *only* the $dE_{\text{GW}}/dJ_{\text{GW}}$ information. Clearly, taking into account information from *both* $dE_{\text{GW}}/dJ_{\text{GW}}$ and f_2 leads to noticeably smaller credible intervals.

As we have argued before, we use 68% contours to approximate the distribution where it has meaningful weight. We agree with the referee that directly comparing 95% and 68% CIs can be misleading. However, in this figure, what is important is the comparison between the solid and dashed 68% CIs (those with the combined slope + f_2 measurement to those with only the f_2 measurement), which illustrates that the joint measurement is more constraining than the measurement of f_2 alone. The 95% contours of this figure are shown only for ease of comparison with Fig. 1 in the manuscript, which uses a different range of n and p values in the EOS figure

(left panel) as compared to Fig. 4. Since we believe that the reader may wish to compare the two figures, we feel it is best to leave the 95% contours on this figure.

Referee: Other comments: Is the definition of f_{GW} in line 142 correct? (pg 4 third line before the end)

Our response: Line 142 is not the definition of f_{GW} , but an equality that holds for a system with only an $\ell = m = 2$ deformation. The definition of the instantaneous gravitational-wave frequency as a derivative of the gravitational wave phase is given in the appendix. Line 142 is however a correct equality that holds in the case of a rotating system with an $\ell = m = 2$ deformation (in geometric units), as the derivation in the appendix shows.

Reviewers #3 (Remarks to the Author):

Referee: To the kind attention of Nature Communication editors and of the authors of the manuscript "Listening to the long ringdown: a novel way to pinpoint the equation of state in neutron-star cores".

The paper deals with an interesting and timely topic. The connection between GWs and the EOS of nuclear matter is certainly a very interesting and broad subject. The relation between the evolution of the energy and angular momentum emission in GWs and the NS properties presented in the paper is valuable. However, I have several concerns about the paper in its present version. These deal with the novelty of the work and with the validity of the presented conclusions. Additionally, I have some remarks about the method and about the manuscript presentation. Below I detail my comments.

Our response: We thank the Referee for noting the interest in the research question and the timeliness of our results.

Referee: In this paper, the Authors present a novel way to constrain the EOS of dense matter through the analysis of the GW post-merger signal produced by a remnant that does not collapse to a BH within the first 10-20ms after merger, i.e. of a HMNS remnant. By analyzing the outcome of merger simulations in Numerical Relativity done by using state-of-art codes in the field, they find a correlation between the ratio at which energy and angular momentum are radiated in GWs, and maximum pressure and density that characterize maximal TOV solutions obtained by a certain EOS. By performing mock measurements of post-merger signals, they find an improvement in the capability of setting constraints on the EOS at several times nuclear saturation density.

Our response: We thank the Referee for a succinct and accurate description of our results.

Referee: I have a few concerns about the novelty and the validity of the paper.

* In lines 122-124 it is stated that the temporal behavior of E_{GW} and J_{GW} was never appreciated so far. However, the behavior depicted in the top panel of figure 2 was already presented and commented in some other

works, see e.g. figure 2 of [10.1103/PhysRevD.94.024023] or figure 5 of [10.1103/PhysRevD.102.024087]. In the first one, the linear relation between E_{GW} VS J_{GW} is evident, as well as a possible dependence on the EOS for the post-merger signal. In the latter, it is also visible the fact that other important parameters of the binary (e.g. the NS spins) can have a similar effect on the E_{GW} VS J_{GW} evolution.

Our response: We thank the Referee for pointing out similar previous works, which we believe are important and have not been cited in the previous version of the manuscript; given their importance, we suppress some of the references to accommodate for them in the revised version.

Indeed, Fig. 2 of [10.1103/PhysRevD.94.024023] (Bernuzzi2015) and Fig. 5 of [10.1103/PhysRevD.102.024087] (Chaurasia2020) highlight a behaviour similar to the one we report in the top panel of our Fig. 2. However, we believe that our analysis differs in several important aspects from that reported in the papers above. First, Bernuzzi2015 points out an “approximately linear” relation between the binding energy and the angular momentum and that the slope is approximately constant and proportional to the GW frequency. In our work, on the other hand, we formally show that for the dominant $\ell = m = 2$ mode of the GW signal, the ratio of the radiated energy and angular momentum rates is exactly proportional to the GW frequency and that if the latter is constant (as measured in the late emission from the remnant) then the ratio dE/dJ must have a linear slope. These considerations are unfortunately not present in Bernuzzi2015. Second, our Fig. 2 provides a comprehensive description of the GW losses in a unified way by combining the inspiral and post-merger part. Essential for this to work is to employ a normalization which allows one to appreciate the universal nature of the inspiral as well as the significant differences in the post-merger slope. Finally and most importantly, neither of these works draw the connection between the slope and EOS properties, which is a main result of our systematic work.

We have modified the manuscript in the following way to include the references to previous works:

Here, we instead focus on the rates at which energy and angular momentum are radiated by the HMNS when it has reached a quasi-stationary equilibrium at about 10 ms after the merger (see also [Bernuzzi2015, Chaurasia2020

]).

and:

More importantly, it is remarkable that in the latter part of the signal, i.e., in the long ringdown, the normalized radiative losses in E_{GW} and J_{GW} are linearly related, as first noted in [Bernuzzi2015b, Chaurasia2020] and clearly shown in the inset reporting a magnification of the long ringdown.

Referee: Refs. [30-31] are actually indicated as the first place where the long-term behavior of f_{GW} was discussed, but my understanding is that also the top panel of fig.2 was already presented there.

Our response: Unfortunately, we cannot follow Referee’s comment. Our refs. [30-31] are not related to the long-term behaviour of f_{GW} .

Referee: The true novelty of this work seems to me the correlation represented by eq 1, where the slope and the EOS properties are directly connected. However, empirical/analytical models of the post-merger GW signal, as for example the one presented in ref [32] or in [10.1088/0264-9381/33/8/085003] or in [10.1103/PhysRevD.100.043005], were also calibrated against the outcome of Numerical Relativity simulations. So they likely include the correct evolution of E_{GW} and J_{GW} as a function of time, and thus of the ratio of their derivatives. So, despite the fact that they do not contain an explicit relation between the value of the E_{GW} VS J_{GW} slope and the EOS-dependent parameters, it seems to me that an analysis performed with those post-merger models should implicitly include that information. What do the authors think? In other words, in which sense should their approach be different from or better than a post-merger GW model that reproduces the same feature in inferring the EOS properties, as it was done for example in [10.1103/PhysRevD.105.104019] or [10.1103/PhysRevD.100.104029] or [10.1103/PhysRevLett.120.031102]?

Our response: As pointed out by the Referee, the previous works mentioned “...do not contain an explicit relation between the value of the E_{GW} VS J_{GW} slope and the EOS-dependent parameters.” This is not a detail as we are here suggesting a novel systematic approach on how to use information that may have been available before but has not yet been used as an effective tool to explore the EOSs at the highest densities. This is analogous

to pointing out a newly discovered universal relation – if it is universal, then it must have been present in previous studies but never noticed or exploited. The novelty of the present work is the identification of the correlation of the slope with n_{TOV} and p_{TOV} , and an illustration of how to use it to further constrain the EOS. In addition, another important and novel aspect of our work is a mathematically well-defined and physically sound approach to study the possible parameter space of model-independent EOSs. Such an approach, based on a PCA, has not been employed before and will represent an effective tool to be employed by numerous studies in the future.

We believe that both of these aspects have significant merits and marked features of novelty and impact.

Referee: Concerning the mock measurement illustrated in lines 200-216 and in the “Performing the mock measurement” section in the SM, my understanding from the text is that the authors use a multivariate Gaussian distribution for f_2 and for the normalized dE/dJ true value, assigning in both cases a 8% uncertainty in a way which looks very arbitrary to me. How realistic is it to assume the same uncertainty for f_2 (the peak of the GW PSD) and for the slope of J_{GW} VS E_{GW} (which must be determined from the GW signal, together with their values at merger, since the correlation was studied for the normalized ratio)? Wouldn’t it be more correct to simulate a certain signal, with a certain SNR, and extract from there all the relevant quantities to have more meaningful estimate of the uncertainties? Alternatively, how much does the result depend on the assumption that the slope of the normalized J_{GW} VS E_{GW} relation have the same uncertainty as the f_2 peak?

Our response: The Referee is correct that such an approach would be more accurate and such an analysis is ongoing as a follow-up to the current work. However, the current estimates of the uncertainties used in Fig. 3 for both measurements are consistent with the current literature (for the errors on f_2 expected from 3rd generation detectors) as well as our analysis of the uncertainties associated with extracting the slope dE/dJ . That is, our literature search and analysis indicate that these uncertainties indeed can be expected to have a similar magnitude. We used equal values just for convenience in the figure. In practice, we found the standard deviation of the measurement of dE/dJ to be about 3%, so we have been conservative

in Fig. 4.

We have added a clarifying footnote about this point in the discussion of Fig. 4

Considering that the standard deviation of the measurement of $d\hat{E}_{\text{GW}}/d\hat{J}_{\text{GW}}$ has been found to be of about 3%, our error estimates in Fig. 4 have been rather conservative. Only when using a distinct and more extensive analysis taking into account realistic GW-signal-processing pipelines will it be possible to set less conservative error estimates.

Referee: The relationship between the normalized slope and the EOS-dependent NS parameters is certainly interesting. It has the potential to be useful in adding a new piece of information in future GW analysis. However, the significance that I see is possibly more limited than the one presented by the authors.

* Some of the statements that are supposed to stress its potential significance seem to me a bit overrated. For example, I do not agree with lines 185-187. The density related to low mass NS refer to the inspiral phase. However, after merger, the density inside the HMNS increases and is probably a significant fraction of n_{TOV} . So, it is not so surprising to me that a relation deduced from the post-merger signal set constraints on the high density part of the EOS.

Our response: We agree with the Referee that this sentence was not optimally worded. Indeed, it is not surprising that the long ringdown probes higher densities than are probed in the inspiral. However, what is striking is that it is not only the “high-density part of the EOS” that is correlated with the slope that we measure, but the “highest” (TOV) densities and pressures, which lie even beyond those probed in the long ringdown. Again, we believe this is not a detail or a small difference.

We have reworded this sentence to highlight this point:

It is straightforward to appreciate from the six panels that the correlation is strong and we find it quite striking that measuring the long-ringdown slope of a low-mass NS can provide precise information on the properties of matter at the highest densities and pressures realized in nature and which are well above those probed in the merger remnant.

Referee: Moreover, according to lines 213-215, the improvement due to

the inclusion of the novel correlation is “considerable”, with respect to the case in which only f_2 was considered. Honestly, looking at figure 4, the improvement looks rather limited to me (20% level). Moreover, this was obtained by considering a rather small and equal errors on f_2 and on the normalized E_{GW} VS J_{GW} slope. How would perform an analysis that uses only the normalized slope, i.e. without f_2 ? How do the results change with respect to the error on the slope?

Our response: We agree with the Referee and we have removed the adjective “considerable” in the revised version of the manuscript.

Referee: In addition to providing further constraints on the functional relations $P = P(n)$ and $M = M(R)$, it seems to me that, in the way it was presented, this relation has little impact on our understanding on the physics that regulates the EOS. Stated differently, to which extend this relation has the potential of distinguishing between different nuclear models, e.g. models that feature a strong phase transition, a smooth phase transition, or no phase transition to quark matter? Such a potential could be very interesting, while a further generic constraint would be very similar to the outcome of several previous works on EOS-constraints from the post-merger signal, maybe based on models that reproduced internally the same relation stressed in this work.

Our response: As we state in the text: “... by construction, our global sample of EOSs, and hence also the golden EOSs, do not contain strong phase transitions...”. In the same paragraph we note “...we focus our attention on smooth EOSs and to build an understanding of their phenomenology, leaving the exploration of EOSs with phase transitions to a subsequent work ...”. Hence, while we appreciate and share the Referee’s interest to use the slope to distinguish models that feature a phase transition, this aspect cannot be addressed in this initial work. The current manuscript forms the basis for future studies and our future analysis will indeed concentrate on addressing the interesting point raised by the Referee.

We have added modified in the text in the conclusions to emphasize the possibility of phase transitions:

The preliminary study carried out here can be improved in a number of ways, e.g. by estimating the impact that large spins, strong magnetic fields,

neutrino emission, strong first-order phase transitions, and temperature-dependent EOSs have on the long-ringdown slope.

Referee: The method is overall fine and the analysis method looks correct. However, also in this case, some potential flaws or weaknesses are present.

* At line 328-329, the authors write about an “intermediate resolution”. What does intermediate mean here? Between what? I completely understand that a large parameter space does not allow high resolution simulations, but 300m looks rather coarse to me. Did the authors try to estimate the potential impact of resolution on their results?

Our response: A resolution of 300 m is surely not very high and a resolution of 200 m may be seen as a more conservative choice. However, these numbers alone do not mean much if not accompanied by a consideration on the truncation error of the numerical methods employed. In our case, the FIL code adopts a fourth-order scheme that at 300 m provides a truncation error that is 30% smaller than the corresponding error at 200 m for a 2nd-order scheme, as often employed in the literature [this estimate is considering an effective third-order convergence which has been demonstrated for FIL in Ref. [44]]. Hence, because the purpose of this work is to explore a reasonable part of the EOS and BNS parameter space, while keeping the computational costs affordable, we believe that a resolution of 300 m is adequate.

However, in response to the Referee’s comment, we have now performed additional simulations with higher (205 m) and lower (395 m) resolutions. In contrast to the EOS dependence, we find that the slope dE/dJ is extremely insensitive to the resolution. Even at the low resolution, the slope values we find are essentially identical to those for higher resolution results. The reason for this is that the post-merger waveform is essentially quadrupolar and dominated by the global dynamics of the merger which is only marginally influenced by the fine details inside the remnant.

In the revised version we have removed the adjective “intermediate” and extended the subsection “Merger Simulations and GW Analysis” with a discussion on the robustness of the slope with respect to the resolution, providing an additional figure:

Next, we demonstrate the robustness of our long-ringdown slope computation with respect to the grid resolution used in the numeric simulation. To

this scope, we performed, in addition to our standard resolution (295 m), also simulations with higher (205 m) and lower (394 m) resolutions. The results of these simulations are summarized in Fig. 8, which highlights how the slope is essentially insensitive to the resolution and that even simulations with low resolutions result in slope values that are well within the uncertainty of the fit (± 0.1 , see discussion below). More specifically, we measure slopes of $\{2.99 \pm 0.06, 3.06 \pm 0.07, 3.02 \pm 0.06\}$ for grid-resolutions of $\{205 \text{ m}, 295 \text{ m}, 394 \text{ m}\}$, respectively. More importantly, the measure of the slope at different resolutions displays a much smaller variance than the equivalent measure of f_{rd} (see lower panel of Fig. 8). The somewhat surprising robustness of the long-ringdown slope with resolution can be simply explained by the fact that the post-merger waveform is dominated by the large-scale $\ell = 2, m = 2$ deformations of the merger remnant, which are only weakly influenced by the small-scale features within the remnant.

Referee: In the conclusions, it is written that the correlation is robust against:

- the chirp mass; - unequal mass binaries; - temperature dependent EOSs;
- different values of the thermal index;

supported by the paragraph "On the robustness of the correlation" in the SM.

It seems to me that this is not really supported for all variables. In particular:

- The tests on M_{chirp} indicate that the impact of M_{chirp} is small. Moreover, since M_{chirp} will be determined with high accuracy for GW signals with high SNR, even a residual small dependence is not a big issue. So, this is fine.

- Moving to the unequal mass cases, I do not understand why the $q = 0.7$ and $q = 0.85$ cases are separately discussed in lines 395-410 and why in the conclusions it is written that the "novel correlation" applies also to them. To my understanding, also such cases were included in the determination of the beta coefficients of eq (1) and they were present in fig 3. So, why do the authors check the robustness of the correlation against cases used to compute the correlation itself? It seems to me that in the SM the authors simply show more details about the unequal mass mergers.

Our response: We agree with the Referee, this is indeed just a presentation

of the unequal-mass results we did not show in the main text. These results were of course used to construct the corresponding correlations presented in the main text. We rewrote the corresponding discussion about the unequal-mass cases:

Next, we show in Fig. 10 results analogue to Fig. 8, but for mass ratios $q = 0.85$ and $q = 0.7$, which complement the information shown in Fig. 3. Note that also the unequal-mass binaries show a clear linear correlation in the radiated energy and angular momentum during the long ringdown and that different mass ratios lead to slightly different slopes.

Referee:

- The thermal index analysis is analogue to the chirp mass analysis, so it is ok.

- The conclusion about the temperature dependent EOS is not correct. The only think that the SM shows is that in simulations employing temperature-dependent EOSs the evolution of E_{GW} VS J_{GW} is qualitatively similar to the one obtained with the thermal index EOS (which was already implied by the simulations showed in [31], for example). But this does not imply the quantitative (or even qualitative) validity of the correlation found by employing EOSs characterized by the thermal index also for the other EOSs. How do the correlation presented in eq (1) and table 2 applies to the results obtained with the HS-DD2 and the V-QCD EOS? Such a test could be beneficial.

Our response: The T -dependent EOSs have been presented to confirm that the linear relation between E_{GW} and J_{GW} is generic and not an artefact of the simplified gamma-law parametrization for the temperature dependence. We find that the slope is not sensitive to the temperature dependence of the EOS. This we demonstrate in Fig. 10 for different fixed values of $\Gamma_{th} = 1.5 - 2.0$, which cover well the spread of the corresponding thermal index of microscopic models such as V-QCD (see Fig.6 in 2112.12157). Since the slopes determined for fixed Γ_{th} do not depend on the specific value used, we expect this also to be the case for microscopic temperature dependence.

In the revised version of the manuscript we have also added the results for HS-DD2 and V-QCD in the correlation plots in Fig. 3. As expected, we observe the V-QCD EOS is in good agreement with the fit obtained using our methodology. The HS-DD2 EOS lies well outside the fit, but this is

again unsurprising as HS-DD2 EOS is in tension with the tidal deformability constraint coming from GW170817.

We have added the following sentence to the caption of Figure 3:

Also shown are data points for two microscopic EOS models, one of which (HS-DD2) is disfavored by astrophysical data.

Referee:

* Finally, at line 175, if I understand correctly, a correlation like eq 1 is also applied to f_2 to test the impact of a single f_2 VS a joint (f_2 +slope) analysis. Is it correct? If so, why such a correlation is used also for f_2 ? Would it be possible that other functional relations work better for f_2 ? Are there other correlations involving f_2 in the literature? Moreover, I do not understand the “respectively” in the text. What is the second term?

Our response: The Referee is correct that a similar correlation as Eq. (1) is also applied to the f_2 data in our analysis. We used a bilinear fit for simplicity, and we did find that it fits the data well. We chose to use a fit to our own golden EOSs rather than any existing fits involving f_2 in order to have a consistent error estimation between the analysis of the slope and f_2 . If we had simply reused an existing fit that had been optimized on some other set of EOSs, then the statistical meaning of our six golden EOSs would be lost.

We agree that our choice of the word “respectively” was not clear. We revised the text as follows to clarify that we have contrasted the quality of the fits to the slope and f_2 :

In particular, we have measured the Pearson-correlation coefficients

$$r(X, Y) := \text{cov}(X, Y) / (\sigma_X \sigma_Y)$$

between the data from our golden EOSs to be

$$r(dE_{\text{GW}}/dJ_{\text{GW}}, P_{\text{TOV}}) = 0.877$$

and

$$r(dE_{\text{GW}}/dJ_{\text{GW}}, n_{\text{TOV}}) = 0.917$$

in the case of the slope, and $r(f_2, P_{\text{TOV}}) = 0.792$ and $r(f_2, n_{\text{TOV}}) = 0.865$ in the case of f_2 , thus indicating that there is a strong correlation in both cases, but also that this is stronger for the long-ringdown frequency.

Moreover, we have added the following sentence to clarify that a second bilinear model was performed, and to comment on the quality of the fit:

For this analysis, we use a second bilinear model to fit the f_2 data, whose quality is found to be as good as the above fit to $d\hat{E}_{\text{GW}}/d\hat{J}_{\text{GW}}$.

Referee: The text is overall well written. However, the flow is not always optimal. And some sentences are unclear or partially incorrect.

* At line 47, it is stated that the remnant of GW170817 collapsed to a BH 1 second after merger. While I think there is a general consensus that GW170817 did not promptly collapsed to a BH, the fact that it survived for 1 sec is not so well established. The analysis reported in [5] is valuable, but requires a certain number of assumptions. Other analysis seem to point to shorter lifetime (for example, a long lived remnant could have produced a blue kilonova only or a much brighter MHD-powered kilonova, see e.g. [10.1093/mnras/stx1987] or [10.1093/mnras/stu802]. Due to the large number of uncertainties in our understanding, I would be more cautious or generic about this point.

Our response: While we agree with the Referee that the collapse time of GW170817 is uncertain and that the estimate in [5] is the result of a number of assumptions, we note that similar estimates have been obtained also in other works and more recently by Murguia-Berthier et al. (ApJ, 629, 870 2021), where a collapse time of 1-1.7 s has been estimated. Again, we have not cited this work because of the reached number of allowed references, which, unfortunately, makes it difficult to include the additional references 10.1093/mnras/stx1987 and 10.1093/mnras/stu802.

Having said that, the revised text is more cautious about the estimate. We have modified the text to the following:

Finally, the analysis of the electromagnetic counterpart associated with GW170817 provided convincing evidence for the formation of a hypermassive neutron star (HMNS) that collapsed into a black hole over a timescale that, under a number of assumptions, has been estimated to be of approximately one second after the merger [Gill2019, Murguia-Berthier2020]

Referee: The first sentence after line 200 (“Having pointed out...”) is unclear to me. It seems that there is a missing piece implied by “between”.

Our response: We have improved the sentence in the revised version (already quoted above):

Having pointed out a novel correlation between the properties of the long-term GW signal and the properties of the EOS at the *highest density*, we now take our analysis a step further and show how our results, in conjunction with a future post-merger GW detection, can be used to constrain the EOS at *all densities*.

Referee: I am confused by the line of reasoning contained in lines 149-176. A) In 149-152 it is written that for a signal with a high enough SNR the slope between E_{GW} and J_{GW} can be measured accurately. But more details about the way to extract such a slope, as well as the uncertainties on it, are not provided.

Our response: The point we are making in lines 149-176 is rather straightforward: the measure of the slope depends on an accurate measure of the radiated GW energy and angular momentum). The latter are computed from the measured GW strain and is therefore clear that the higher the SNR, the better will be to measure the strain and so the slope.

We have also added an extended discussion on how we compute the slope from our simulations at the end of the subsection “Merger Simulations and GW Analysis” in the Methods (see also reply to Referee #1):

Our final discussion on the methods employed in our analysis is focussed on the accuracy of the slope extraction from the numerical simulations. This operation requires first to identify the optimal time-range for the linear least-squares fit of $E_{\text{GW}}(J_{\text{GW}}(t))$, whose starting and final times are determined by minimizing the variance of the linear fit. More specifically, we first compute the slope and its variance for a number of different starting times $t_{\text{in}} - t_{\text{mer}} \in [1 - 10]$ ms using a fixed value for the final time $t_{\text{fin}} - t_{\text{mer}} = 15$ ms. In this way, we found that a starting time of $t_{\text{in}} - t_{\text{mer}} = 1$ ms results in an approximate variance of ± 0.1 for the slope. Larger values for the starting time, e.g. 5 ms or 10 ms, result in significantly larger variances of ± 0.7 and ± 3.1 , respectively. Next, we compute the slope and its variance for various values of the final time $t_{\text{fin}} - t_{\text{mer}} \in [2 - 25]$ ms while keeping the starting time fixed at $t_{\text{in}} - t_{\text{mer}} = 1$ ms. In this way, we found that the variance saturates at $t_{\text{fin}} - t_{\text{mer}} \approx 15$ ms to values similar to those obtained by varying

the starting time. Increasing the final time of the fit does not lead to further improvement of the fit quality and this is because it becomes increasingly difficult to accurately compute the small changes in E_{GW} and J_{GW} at times beyond $t - t_{\text{mer}} \gtrsim 15$ ms, where the amplitude of the GW signal becomes very small.

Referee: B) In 156-159 it is written that f_{rd} and the slope between E_{GW} and J_{GW} have the same value. But this is not true: as it is written at line 160 and one can verify also later in the text, the two quantities are conceptually analogue, but different. For example, according to the simulation results, f_{rd} is much closer to f_2 than the normalized slope.

Our response: We thank the Referee for this important remark. Indeed the asymptotic frequency f_{rd} and the frequency obtained from the normalized slope are conceptually different and are not in general equal. Indeed, as we say in the main text and show in Eq. (11) the normalized slope and the f_{rd} coincide if the assumption that the signal is dominated by the $\ell = m = 2$ mode is satisfied.

We have corrected the text following the Referees comment clarifying the conceptual difference of f_{rd} and the frequency extracted from dE/dJ :

This panel shows in fact that during the long ringdown f_{GW} also asymptotes to an essentially constant value, $f_{\text{GW}} = f_{rd} \simeq \text{const.}$ (see also [Bernuzzi2015b] where this behaviour was first mentioned), and—assuming that the signal is dominated by the $m = l = 2$ mode—this is the same value that can be deduced from the slope between E_{GW} and J_{GW} .

Referee: C) The statement that the slope is much robust to fit than f_{GW} (lines 160-161) is not obvious: this depends on the uncertainties that we can associate to the quantities we have to fit. If E_{GW} and J_{GW} have significantly larger uncertainties than f_{GW} , this cannot be the case. Moreover, it is unclear where this is discussed more in the SM.

Our response: We agree with the Referee that an accurate assessment of the robustness of the determination of the slope under realistic conditions of a GW detection depends on the uncertainties of E_{GW} , J_{GW} , and f_{GW} . An analysis of this type is of course both interesting and important, but goes beyond the scope of this initial work where the novel strategy is outlined.

Indeed, we will soon start a new study that includes a full estimation of these uncertainties and the use of realistic GW-detection pipelines.

Within the scope of the current analysis we have assessed whether it is easier to measure the slope of $E_{\text{GW}}/J_{\text{GW}}$ using a linear fit to our simulation data, or to calculate the average of $f_{\text{GW}} \approx \dot{E}_{\text{GW}}/\dot{J}_{\text{GW}}$ from the simulation. In this way, we have found that the former is more robust because while each data point has a small uncertainty on the y (normalized E_{GW}) axis, there are many points spread out over a large range on the x (normalized J_{GW}) axis. This is to be contrasted with the calculation of the average of f_{GW} , which does not benefit from the extended range on the x axis.

In other words, for a linear fit $y_i = b_0 + b_1 * x_i$, assuming normally distributed residuals of the fit $\eta_i = b_0 + b_1 * x_i - y_i$ with standard deviation σ , the variance of the parameter b_1 is given by $(\sigma^2 / \sum((x_i - \langle x \rangle)^2)) < \sigma^2$. In our case, the uncertainties on E_{GW} , J_{GW} and f_{GW} are such that the variance in b_1 is comparatively small, so that calculating the slope has a smaller error.

We have added the following footnote to the manuscript to address this point:

This is because the variance of the slope in a linear fit of data $\{(x_i, y_i)\}$ is suppressed by an additional factor $\sum_i (x_i - \langle x \rangle)^2$ when compared to the residual variance of the fit.

Furthermore, we thank the referee for pointing out that we did not discuss this more in the supplemental material despite our remark. We have now removed this spurious remark from the manuscript.

Referee: D) How are the “extrapolated slopes” of figure 2 determined?

Our response: This is an important point which we indeed should have discussed in more detail. We perform a linear least squares fit of $E(J(t))$ starting at 1 ms until the final simulation time (~ 30 ms) after the merger. The slope values reported in Table 1 are the slopes determined via these fits. The light colored lines in Fig. 2 show these linear fits and are, as a guide for the eye, extended beyond the maximal value of J reached in the simulations. Our choices for the starting and final times for the fit are determined by minimizing its variance. For this we have computed the slope and its variance for a number of different starting times (1-10 ms)

for fixed end time 15 ms after merger. From this we find that a starting time of 1 ms results in a variance of approximately ± 0.1 . Larger values, e.g., 5 ms or 10 ms result in significantly larger variances of ± 0.7 and ± 3.1 , respectively. Then we computed the slope and its variance for various values of the final time (2-25 ms) while keeping the starting time fixed at 1 ms. From this we find that the variance saturates at around 15 ms to the one obtained from varying the starting point, i.e., ± 0.1 . This is the ultimate accuracy of our slope determination. End times larger than 15 ms do not decrease the variance further (see Figure 2 (bottom)). This is because it becomes numerically increasingly difficult to accurately compute the very small increases in E and J from the simulation, since the GW amplitude becomes typically small 15 ms after the merger, leading to an approximate “0/0” situation for $dE_{\text{GW}}/dJ_{\text{GW}}$. What discussed above can be appreciated in the attached Fig. 2, which reports on the top the value of the slope (and its variance) when varying the starting time (in milliseconds) of the fit with fixed a end-time of 20 ms. Similarly, the bottom panel of Fig. 2 shows the same quantities when varying the end-time of the fit and starting from 1 ms after the merger.

We added an extended discussion on this at the end of the subsection “Merger Simulations and GW Analysis” in the Methods (see also reply to Referee #1):

Our final discussion on the methods employed in our analysis is focussed on the accuracy of the slope extraction from the numerical simulations. This operation requires first to identify the optimal time-range for the linear least-squares fit of $E_{\text{GW}}(J_{\text{GW}}(t))$, whose starting and final times are determined by minimizing the variance of the linear fit. More specifically, we first compute the slope and its variance for a number of different starting times $t_{\text{in}} - t_{\text{mer}} \in [1 - 10]$ ms using a fixed value for the final time $t_{\text{fin}} - t_{\text{mer}} = 15$ ms. In this way, we found that a starting time of $t_{\text{in}} - t_{\text{mer}} = 1$ ms results in an approximate variance of ± 0.1 for the slope. Larger values for the starting time, e.g. 5 ms or 10 ms, result in significantly larger variances of ± 0.7 and ± 3.1 , respectively. Next, we compute the slope and its variance for various values of the final time $t_{\text{fin}} - t_{\text{mer}} \in [2 - 25]$ ms while keeping the starting time fixed at $t_{\text{in}} - t_{\text{mer}} = 1$ ms. In this way, we found that the variance saturates at $t_{\text{fin}} - t_{\text{mer}} \approx 15$ ms to values similar to those obtained by varying the starting time. Increasing the final time of the fit does not lead to further

Figure 2: Sensitivity of the calculation of the slope to the details of the time windowing. *Top panel:* value of the slope (and its variance) when varying the starting time (in milliseconds) of the fit with a fixed end-time of 20 ms. *Bottom panel:* the same but when varying the end-time of the fit and starting from 1 ms after the merger.

improvement of the fit quality and this is because it becomes increasingly difficult to accurately compute the small changes in E_{GW} and J_{GW} at times beyond $t - t_{\text{mer}} \gtrsim 15$ ms, where the amplitude of the GW signal becomes very small.

Referee: E) How are the Pearson-correlation coefficient quoted in lines 171-176 computed? It is unclear from which analysis they are deduced.

Our response: The Pearson coefficient is defined as

$$p(X, Y) = \frac{\text{cov}(X, Y)}{\sigma_X \sigma_Y},$$

where σ_X and σ_Y are the standard deviation of the X and Y data respectively. It is a measure of the linear correlation between the data X and Y . As this is a coefficient characterizing the data itself and not the fits, it is calculated using the data points in Fig. 3.

Since this is a point that has been discussed also with Referee #1 and #2, we note the new discussion that we have added in the text

In particular, we have measured the Pearson-correlation coefficients

$$r(X, Y) := \text{cov}(X, Y) / (\sigma_X \sigma_Y)$$

between the data from our golden EOSs to be

$$r(dE_{\text{GW}}/dJ_{\text{GW}}, P_{\text{TOV}}) = 0.877$$

and

$$r(dE_{\text{GW}}/dJ_{\text{GW}}, n_{\text{TOV}}) = 0.917$$

in the case of the slope, and $r(f_2, P_{\text{TOV}}) = 0.792$ and $r(f_2, n_{\text{TOV}}) = 0.865$ in the case of f_2 , thus indicating that there is a strong correlation in both cases, but also that this is stronger for the long-ringdown frequency.

Referee: Overall, in addition to fixing point B), I think more details are necessary to support the other points.

Our response: We agree with the Referee that this is an important point and we hope that our reply above to concerning the point B) is satisfactory to the Referee.

1 Reviewer #4 (Remarks to the Author):

Referee: The authors propose a new method for connecting measurable quantities from gravitational-wave observations of long post-merger parts of binary-neutron-star mergers and the equation of state of high-density matter, which is not accessible through experiments or observations of stable neutron stars. With a statistical analysis, they also show how the new method improves on previous methods and highlight its applicability to future observations. In detailing the many advantages of the new method, the authors also provide plenty of convincing comments on possible objections a reader might have to its generality and applicability.

The codes used in this work have been thoroughly tested and there is no reason for doubting the results of the simulations, but I checked with the data of our own simulations and found supporting evidence.

This is an important work, which will very likely allow more accurate identification of the properties of the equation of state in future gravitational-wave measurements. And, thus, it is a significant advance in the field.

The presentation of the background, idea, setup and results is very good: clear, detailed, and accessible to a range of physicists with different specializations.

Our response: We thank the Referee #4 for a succinct and accurate summary of our work. We especially thank the Referee for the work put to verify the reproducibility of our results and acknowledging the importance and usefulness of the results.

Referee: Below are some questions and comments the authors may consider.

1) On lines 107, ref. [24] is cited for $\Gamma_{thermal} = 1.75$. However, Figura et al. PRD102 (which is ref. [24]) state that: “we have concluded that a value of $\Gamma_{thermal} \approx 1.7$ best approximates the complete, finite-temperature EOS in binary NS simulations. This value is similar to the standard one employed in numerical simulations so far (i.e., $\Gamma_{thermal} = 1.75$ – 1.80), but also importantly lower.”

It is customary to use $\Gamma_{thermal} = 1.75$ in BNS simulations and there is no problem with that, but I recommend the authors to cite a more appropriate reference than [24].

Our response: We agree and thank the Referee for pointing out this inaccuracy, which we have corrected in the revised version, where we state:

...While this is an approximation, it does not affect the properties of the correlation and we have adopted an adiabatic-index value of $\Gamma_{\text{th}} = 1.75$, which is close to the optimal value ($\Gamma_{\text{th}} \approx 1.7$) suggested in [25] and on average mimics well the temperature dependence of microscopic constructions [26]...

Referee: 2) In the sentence on lines 184-7, can the second statement (“and we find it quite remarkable...”) also be deduced directly from Fig. 3? If so, the authors may clarify how.

Our response: The logic here is that the correlations shown in Fig. 3 are between the maximum (TOV) neutron star densities/pressures and the GW ringdown slopes. The long ringdown slopes are deduced from binary mergers that feature initial binary constituent masses that are realistic and consistent with the observations, e.g., $1.3 - 1.4 M_{\odot}$. In this sense, they are significantly smaller than M_{TOV} (e.g., $2 - 2.5 M_{\odot}$) and therefore they probe densities/pressures much smaller than the maximum-mass stars.

As remarked above in the response to Referee #3, we have improved this sentence stress that it is the relation to the TOV quantities that is striking:

It is straightforward to appreciate from the six panels that the correlation is strong and we find it quite striking that measuring the long-ringdown slope of a low-mass NS can provide precise information on the properties of matter at the highest densities and pressures realized in nature and which are well above those probed in the merger remnant.

Referee: Finally, a very small lexical comment: On line 163, maybe “transparent” is not the right term for describing the lines in paler colors.

Our response: We thank the Referee for the comment and we use “transparent” because paler colours are simply bright colours with an increased level of transparency. That said, in the revised version we have replaced the adjective “transparent” by “thin”, which hopefully be more clear.

Reviewers # 1 and # 2 (Remarks to the Author):

Referees: Report: The authors have addressed all of our comments and have supplemented the text where appropriate. However, we still believe that the same statements should be made more carefully, taking into account the conditions under which the study was carried out, in particular the fact that the analysis was carried out considering EOS within the 68% CI.

Our response: We thank the Referees for acknowledging that all their comments have been appropriately addressed and for pointing out the additional minor improvements we are happy to implement in the new version of the manuscript.

Referees: 1. The authors write “In principle, one could generalize this analysis by considering additional uncorrelated variables to the four we have chosen, or even attempt to identify the optimal set of uncorrelated variables, but this is beyond the scope of this work.” The question is precisely whether the chosen set of uncorrelated variables is the optimal one, and whether a different set would not lead to a “golden” EOS with different behavior. Even if this is beyond the scope of the present study, the need to identify the optimal set of uncorrelated variables should be made more explicit. Besides the authors’ option of considering a 68% CI does not leave much room for extra uncertainties.

Our response: We thank the Referees for this intriguing and challenging question. Determining the optimal choice of EOSs is far from trivial and even the bare definition may require to take into account also the response properties of the GW detector, since what is optimal for one detector may not be optimal for another. So, while addressing this point is interesting per-se, it clearly requires a study of its own, which is beyond the scope of our paper. More importantly, it is quite relevant for our results.

We have now made this issue more explicit by adding the following sentence to the final paragraph of the main text:

Additionally, the set of neutron-star parameters used in the principal-component analysis could be extended and optimized.

In order to address the second part of the Referee’s question, we want to point out that the general merger dynamics and the pattern of GW emission

and therefore the appearance of a long ringdown phase where E_{GW} and J_{GW} are linearly related does not depend on the fine details of the EOS. In this sense we expect different golden EOSs will lead to quantitatively different, but qualitatively equivalent results. In fact, this is supported by our Fig. 12, where the same generic behaviour is shown for two EOSs that are entirely different from our golden ones. Furthermore, the linear slope behaviour has been also pointed out previously by other authors [28,29] for different EOS models and even confirmed by independent simulations by Referee 4. Our analysis, together with previous results in the literature and independent checks by Referee 4, provide strong evidence that the long ringdown phase is a generic feature of any EOS model.

Referees: 2. The authors write “The Referee correctly points that different theoretical treatments may lead to differences in $R_{1.4}$. This is, however, not a valid point of criticism towards our work as our EOSs arise from a model-independent prior which has been empirically constrained to this range. Deviation of this range would necessarily lead to being in tension with the LIGO tidal-deformability measurement and/or the existence of two-solar-mass stars.”

We agree with the statement that the EoS should cover the region defined by NICER and LIGO in M-R space, but it has been shown that a non-unified crustal EOS can introduce uncertainties in the radius estimate, and therefore the model-independent EOS determined by the observational constraints is affected by the model-dependent crust (BPS crust), see e.g. Davis et al. 2406.14906 [astro-ph.HE] and references therein.

Our response: We thank the referee for pointing out this additional uncertainty arising from the crustal EOS. (We note that the work cited by the referee appeared after we submitted our manuscript).

We have now modified the following sentence in final paragraph of the main text pointing out this possible additional source of uncertainty:

The preliminary study carried out here can be improved in a number of ways, e.g., by estimating the impact that large spins, strong magnetic fields, neutrino emission, strong first-order phase transitions, and temperature-dependent EOSs and more generic treatment of the crust and sub-saturation density matter have on the long-ringdown slope.

In addition, we have added the following footnote to the Methods section:

Note that the uncertainty associated with the crustal EOS has been recently discussed in [2406.14906].

Referees: 3. The authors write: “In particular, choosing instead the 95% contour would result in a selection of more exotic EOSs, that is, EOSs that have significantly lower posterior likelihood, without impacting the overall results of the our analysis.” It is not clear how the authors can say “without affecting our analysis” as this has not been tested. It is also not clear what is meant by exotic. According to the authors’ interpretation, it seems to be equivalent to being outside the 68% CI, and therefore SFHo is an exotic EOS. However, one would not expect the samples to be analyzed and interpreted as “exotic cases” after the CI has been defined.

According to the methodology presented, SFHo is an exotic EOS, although by construction it satisfies nuclear properties - properties not imposed in the present approach - in addition to observational constraints. One would not say that it is an exotic EOS. The low-density constraints imposed in the present study are those introduced by the beta-equilibrium EOS constructed in Hebel et al. 2013 from the neutron matter EOS determined in a chEFT approach and an effective description of symmetric matter. The band defined by the soft and hard EoS in the $0.5 - 1.1 n_0$ was considered to define a 90% CI. It is difficult to accept that there is more information on nuclear matter at and below the saturation density with the new method introduced in the present work than the one used to determine SFHo. Again, the choice of considering only 68% CI is quite strict to allow for possible uncertainties in the low density EOS.

The new method proposed to infer the behavior of high-density baryonic matter from the knowledge of the long post-merge is certainly important, and as the authors point out, this is a preliminary study that can be improved in a number of ways in the future, including those mentioned above.

Our response: Firstly, we agree with the referee that the choice of word “exotic” was not optimal. We have now removed it. In addition we have changed the wording to emphasise that if different CI would be chosen, this would merely change the way we characterise the underlying distribution, not the underlying distribution itself. Our new text emphasizes that the

choice does not affect the results to the extent that the principal-component procedure characterises the underlying distribution well.

We have modified the paragraph in question as follows:

It is worth noting that using the corners of the 68% credibility contours for the golden EOS selection is a matter of choice in how we represent the underlying distribution. The variability of the simulations with the golden EOSs can be used as a proxy to approximate the 68%-credible regions that would be obtained if the full GP ensemble was used. In our analysis, this choice of 68% represents a compromise between capturing the extrema of the EOS distribution and assuring a sufficiently high posterior probability for the selected EOSs. In particular, choosing instead the 95% contour would represent the same distribution with a selection of EOSs coming from the tails of the EOS distribution, that is, EOSs that have significantly lower likelihood. This choice does not affect the overall results to the extent that our golden set characterises the features of the underlying distribution.

We also note that we now highlight the possibility of including further uncertainties in the low-density EOS in the final paragraph mentioned in point 2 above.

Reviewers # 3 (Remarks to the Author):

Referee: To the kind attention of the editors and of the authors. I read the authors' answers to my points, as well as of the other referees. First of all, I would like to thank the authors for taking into account my concerns and in answering my questions. In most of the cases, the answers were satisfactory and the changes have, in my opinion, improved the readability of the manuscript. Concerning the degree of novelty, I think that the present version better represents the content development. The fact that E_{GW} and J_{GW} have a linear relation is not the true novelty of the work, while I agree with the authors that the relation with the EOS is the real novelty of the work.

I have a few further (minor) comments, which the authors can find below.

Our response: We thank the Referee for acknowledging that all the comments have been appropriately addressed and for pointing out the additional minor improvements we are happy to implement in the new version of the manuscript.

Referee: A) Concerning my previous comment:

“Referee: Refs. [30-31] are actually indicated as the first place where the long-term behavior of fGW was discussed, but my understanding is that also the top panel of fig.2 was already presented there.”

“ Our response: Unfortunately, we cannot follow Referee's comment. Our refs. [30-31] are not related to the long-term behaviour of fGW.”

In the first manuscript, refs. 30 and 31 were:

[30] Bernuzzi, S., Dietrich, T., Nagar, A.: Modeling the Complete Gravitational Wave Spectrum of Neutron Star Mergers. Phys. Rev. Lett. 115(9), 091101 (2015) <https://doi.org/10.1103/PhysRevLett.115.091101>
arXiv:1504.01764 [gr-qc]

[31] Bernuzzi, S., Radice, D., Ott, C.D., Roberts, L.F., Moesta, P., Galeazzi, F.: How loud are neutron star mergers? Phys. Rev. D 94(2), 024023 (2016) <https://doi.org/10.1103/PhysRevD.94.024023>
arXiv:1512.06397 [gr-qc]

The second one is now reference [28], while reference [30] has now disappeared. My point was simply that these references were already present,

but the present version improves upon the original one with respect to a better location in the text and a more appropriate usage/acknowledgment of previous findings. The present version is thus fine with me.

Our response: We thank the Referee for clarifying this point and are glad that the current version addresses the previous comment.

Referee: B) At line 56 the acronym PSD is introduced. I would define it as: “power spectral density” rather than “spectral power density”.

Our response: We have now changed our wording to “power spectral density” in the third paragraph of the manuscript:

This has important consequences, as a number of studies have shown that the most prominent features of the power spectral density (PSD) of the post-merger signal...

Referee: C) I suggest the authors to read again the caption of figure 7. I cannot see the dashed lines representing f_2 . Moreover, I do not understand in the final sentence “...where and shaded...”.

Our response: We thank the Referee for pointing out what is actually a mistake as dashed lines were used in a previous previous version of the figure. We corrected the sentence accordingly and modified the wording in the part that refers to the shaded regions for the error estimate of f_2 :

In the right panel we mark with solid lines the dominant post-merger frequency f_2 where and shaded areas indicate a 8% relative error estimate.

Referee: D) I thank the authors for providing more runs with different resolutions. I still think that dx 300m is a rather coarse resolution for a BNS merger, but I agree that it is probably enough to extract the dominant GW frequencies with sufficient accuracy. Resolution is crucial, for example, for the remnant lifetime or the ejcta properties, but this is not the point of the paper. In lines 414-417, the authors seem to be surprised by this outcome. However, recent resolution studies (e.g. Zappa et al, MNRAS, Volume 520, Issue 1, 2023) confirmed that. Additionally, if possible, at line 360 I would remove “intermediate”.

Our response: We agree with the Referee that 300 ms is a coarse resolution and we would not use it to make accurate predictions on the merger-remnant lifetime or on the properties of the ejecta. However, we all agree it is sufficient for the for the purpose of this paper. Furthermore, we removed the the word “intermediate” in line 360 (now line 380).

Referee: E) I checked the references and I have a few remarks. At line 107, references [12] and [19-24] should support the fact that the temperature in BNS mergers reaches tens of MeV. But all these references, with the exception of 23 and 24, used an hybrid EOS, in which thermal effects are not accurate or consistent. And even in ref 23 and 24, the focus is on hadron-quark phase transition, i.e. the case not explored in the paper. I suggest the authors to search for works in which the temperature and the thermodynamics conditions in BNS merger are analyzed in a more systematic way, using finite-T EOSs and possibly including also works in which no hadron-quark phase transition is present. While I would avoid references to work that uses hybrid EOS approaches.

Our response: While we appreciate the Referee’s suggestion about making more accurate citations to simulations using realistic and temperature dependent EOSs, we feel the present references are appropriate given the very qualitative nature of the sentence, which refers to “tens of MeV” and not to a precise range of temperature. [We incidentally note that a very old paper measured temperatures of the order of tens of MeV despite using a very simplified $\Gamma = 2$ law (0804.0594)]

Referee: When commenting about the $\ell = 2, m = 1$ instabilities, the authors wrote that such mode could be “dominant ... at later times for highly asymmetric binaries”. However, studies like Radice et al 2016, PhRvD, 94, 064011, suggest that also symmetric BNS can produce it. Could the authors comment on that?

Our response: Indeed, also equal-mass binaries can trigger the $\ell = 2, m = 1$ instability and we are not excluding this in our sentence. The point we make is that, in the case of unequal-mass binaries, the $\ell = 2, m = 1$ deformation is already large right after merger and hence requires less time to become the dominant one. By contrast, in equal-mass binaries, the $\ell = 2, m = 1$

deformation is extremely small and a longer timescale will be needed for it to become dominant. We have modified the text as follows so as to make this point clearer.

However, it is possible at later times that the $\ell = 2, m = 1$ mode will dominate (see, e.g., [52]) both for highly asymmetric binaries (for which the $m = 1$ deformation is quite large right after merger), but also for equal-mass binaries (for which the $m = 1$ asymmetry is initially small but grows steadily). Also for this $\ell = 2, m = 1$ GW mode, the radiated energy and angular momentum will remain linearly related, albeit with a different (smaller) slope.

Referee: F) In table 1 of the SM, f_2 , f_{rd} and dE_{GW}/dJ_{GW} are provided as “EOS properties”. How were they computed? This is presently unclear from the text.

Our response: Our procedure for computing dE_{GW}/dJ_{GW} is explained in detail on pp. 17-18 of the manuscript. We apply an identical procedure to determine f_{rd} , which we mention now explicitly for clarity on line 480:

We apply an analogous procedure to determine f_{rd} .

In addition, we mention now also that the values of f_2 listed in Table 1 correspond to the global maxima of Eq. (12) on line 443:

As done routinely (see, e.g., [9-14]), the dominant post-merger frequency f_2 is then determined by the global maximum of the PSD.

Reviewers # 1 and # 2 (Remarks to the Author):

Referees: We thank the authors for considering all our comments and clarifying the manuscript. We recommend the present study for publication in Nature Communications since it proposes a new method of extracting the high-density baryonic matter EOS from the observation of NS mergers, in particular, from the post-merger GW signals. This is a region of the QCD phase diagram not reachable in the lab, one of the major problems in the field.

Our response: We thank the Referees for recommending publication Nature Communications and for the useful comments that have improved the paper.

Reviewers # 3 (Remarks to the Author):

Referee: To the kind attention of the editors and of the authors. I thanks the authors for taking into account my comments and for answering my questions. However, I am not fully convinced by points C, E and F. Below they can find my further answers.

I thank the authors for amending the figure and the capture. I still do not understand the meaning of “...WHERE AND...” in the final sentence. Could they please rephrase?

Our response: Unfortunately the caption still contained a typo. We corrected the sentence accordingly:

In the right panel we mark with solid lines the dominant post-merger frequency f_2 , where the shaded areas indicate a 8% relative error estimate.

Referee: I am sorry for stressing again this point, but I do not agree with the answer. In my comment, I did not write that the estimate or the results of the present references were wrong or not valuable. I simply wrote that the statement that the temperature in BNS mergers reaches tens of MeV is now supported by works and simulations in which a dedicated analysis is provided and in which the temperature is consistently taken into account. To me, these seem to be the most appropriate references to that sentence. If the authors want to keep in practice the present set of references, I am fine (most of these references are anyway used elsewhere in the manuscript), but I would at least add (or replace) one reference citing a paper in which: - investigation of the thermodynamical conditions was done in a systematic way; - finite-T, composition dependent (and, for consistency with the present work, possibly hadronic) EOSs were employed. I would like to notice that in ref [12] (De Pietri et al PRL 2018), the word “temperature” does not appear at all. I am even doubtful if this reference is appropriate as a description of the “general merger dynamics” (as reported a few lines after), since it focuses on a possible GW emission happening on timescales much longer than the ones explored here. Such a GW-emission is rather controversial and could be due to low resolution artifacts: as the authors admitted, a resolution of dx 300m is not appropriate to model the post-merger phase,

and for the precise modeling of such long-term behavior magnetic field , finite-T, and composition effects are not negligible.

Our response: We respectfully disagree with the Referee and believe there is a general misunderstanding about the message that the sentence in question wishes to deliver. More specifically, the sentence aimed at providing a very generic description of the temperature evolution and was not meant to give a precise temperature estimate, which depends sensitively on the EOS. Importantly, the physically relevant part of the sentence is the first one, which remarks that during and after the merger *shocks* will raise the temperature. We have clarified the sentence so that the issue does not rise. The new text is:

While NSs during the inspiral stage can be described also when neglecting the temperature dependence of the EOS, during and after the merger shock-heating effects lead to non-negligible temperatures inside the merger remnant.

Referee: I thank the authors for better explaining the way in which the quantities appearing in table 1 were computed. This is helpful. But I think I did not express my original point in a clear way. At the moment, f_2 , f_{rd} and dE_{GW}/dJ_{GW} are presented as “EOS and NS properties” (see bold text of the caption and the text after it: “For each EOS, we list...”), at the same level as the maximum NS mass or maximum pressure. But all the quantities coming after the q column refer to (and depend on) BNS system modeled using a certain EOS. So, I find the caption misleading. Could the authors improve on this?

Our response: Indeed, the original caption was not entirely precise and we extended it in the revised version such that it explicitly includes BNS quantities. The caption has now been changed to read:

EOS, NS and BNS properties.